# RandProx: Primal–Dual Optimization Algorithms with Randomized Proximal Updates

**Laurent Condat & Peter Richtárik**
Visual Computing Center
King Abdullah University of Science and Technology (KAUST)
Thuwal, Kingdom of Saudi Arabia
Contact: see `https://lcondat.github.io/`

## Abstract

Proximal splitting algorithms are well suited to solving large-scale nonsmooth optimization problems, in particular those arising in machine learning. We propose a new primal–dual algorithm, in which the dual update is randomized; equivalently, the proximity operator of one of the function in the problem is replaced by a stochastic oracle. For instance, some randomly chosen dual variables, instead of all, are updated at each iteration. Or, the proximity operator of a function is called with some small probability only. A nonsmooth variance-reduction technique is implemented so that the algorithm finds an exact minimizer of the general problem involving smooth and nonsmooth functions, possibly composed with linear operators. We derive linear convergence results in presence of strong convexity; these results are new even in the deterministic case, when our algorithms reverts to the recently proposed Primal–Dual Davis–Yin algorithm. Some randomized algorithms of the literature are also recovered as particular cases (e.g., Point-SAGA). But our randomization technique is general and encompasses many unbiased mechanisms beyond sampling and probabilistic updates, including compression. Since the convergence speed depends on the slowest among the primal and dual contraction mechanisms, the iteration complexity might remain the same when randomness is used. On the other hand, the computation complexity can be significantly reduced. Overall, randomness helps getting faster algorithms. This has long been known for stochastic-gradient-type algorithms, and our work shows that this fully applies in the more general primal–dual setting as well.

## 1 Introduction

Optimization problems arise virtually in all quantitative fields, including machine learning, data science, statistics, and many other areas (Palomar & Eldar, 2009; Sra et al., 2011; Bach et al., 2012; Cevher et al., 2014; Polson et al., 2015; Bubeck, 2015; Glowinski et al., 2016; Chambolle & Pock, 2016; Stathopoulos et al., 2016). In the big data era, they tend to be very high-dimensional, and first-order methods are particularly appropriate to solve them. When a function is smooth, an optimization algorithm typically makes calls to its **gradient**, whereas for a nonsmooth function, its **proximity operator** is called instead. Iterative optimization algorithms making use of proximity operators are called proximal (splitting) algorithms (Parikh & Boyd, 2014). Over the past 10 years or so, primal–dual proximal algorithms have been developed and are well suited for a broad class of large-scale optimization problems involving several functions, possibly composed with linear operators (Combettes & Pesquet, 2010; Boţ et al., 2014; Parikh & Boyd, 2014; Komodakis & Pesquet, 2015; Beck, 2017; Condat et al., 2023a; Combettes & Pesquet, 2021; Condat et al., 2022c).

However, in many situations, these deterministic algorithms are too slow, and this is where **randomized algorithms** come to the rescue; they are variants of the deterministic algorithms with a cheaper iteration complexity, obtained by calling a random subset, instead of all, of the operators or updating a random subset, instead of all, of the variables, at every iteration. Stochastic Gradient Descent (**SGD**)-type methods (Robbins & Monro, 1951; Nemirovski et al., 2009; Bottou, 2012; Gower et al., 2020; Gorbunov et al., 2020; Khaled et al., 2020b) are a prominent example, with the huge success we all know. They consist in replacing a call to the gradient of a function, which

can be itself a sum or expectation of several functions, by a cheaper **stochastic gradient** estimate. By contrast, replacing the proximity operator of a possibly nonsmooth function by a **stochastic proximity operator** estimate is a nearly virgin territory. This is an important challenge, because many functions of practical interest have a proximity operator, which is expensive to compute. We can mention the nuclear norm of matrices, which requires singular value decompositions, indicator functions of sets on which it is difficult to project, or optimal transport costs (Peyré & Cuturi, 2019).

In this paper, we propose RandProx (Algorithm 2), a randomized version of the Primal–Dual Davis–Yin (PDDY) method (Algorithm 1), which is a proximal algorithm proposed recently (Salim et al., 2022b) and further analyzed in Condat et al. (2022c). In RandProx, one proximity operator that appears in the PDDY algorithm is replaced by a stochastic estimate. RandProx is **variance-reduced** (Hanzely & Richtárik, 2019; Gorbunov et al., 2020; Gower et al., 2020); that is, through the use of control variates, the random noise is mitigated and eventually vanishes, so that the algorithm converges to an exact solution, just like its deterministic counterpart. Algorithms with stochastic errors in the computation of proximity operators have been studied, for instance in Combettes & Pesquet (2016), but the errors are typically assumed to decay or some stepsizes are made decaying along the iterations, with a certain rate. By contrast, in variance-reduced algorithms such as RandProx, which has fixed stepsizes, error compensation is automatic.

We analyze RandProx and prove its linear convergence in the strongly convex setting, with additional results in the convex setting; we leave the nonconvex case, which requires different proof techniques, for future work. We mention relationships between our results and related works in the literature throughout the paper. In special cases, RandProx reduces to Point-SAGA (Defazio, 2016), the Stochastic Decoupling Method (Mishchenko & Richtárik, 2019), ProxSkip, SplitSkip and Scaffnew (Mishchenko et al., 2022), and randomized versions of the PAPC (Drori et al., 2015), PDHG (Chambolle & Pock, 2011) and ADMM (Boyd et al., 2011) algorithms. They are all generalized and unified within our new framework. Thus, RandProx paves the way to the design of proximal counterparts of variance-reduced SGD-type algorithms, just like Point-SAGA (Defazio, 2016) is the proximal counterpart of SAGA (Defazio et al., 2014).

## 2 PROBLEM FORMULATION

Let $\mathcal{X}$ and $\mathcal{U}$ be finite-dimensional real Hilbert spaces. We consider the generic convex optimization problem:

$$\text{Find } x^\star \in \arg\min_{x \in \mathcal{X}} \Big( f(x) + g(x) + h(Kx) \Big), \tag{1}$$

where $K : \mathcal{X} \to \mathcal{U}$ is a nonzero linear operator; $f$ is a convex $L_f$-smooth function, for some $L_f > 0$; that is, its gradient $\nabla f$ is $L_f$-Lipschitz continuous (Bauschke & Combettes, 2017, Definition 1.47); and $g : \mathcal{X} \to \mathbb{R} \cup \{+\infty\}$ and $h : \mathcal{U} \to \mathbb{R} \cup \{+\infty\}$ are proper closed convex functions whose proximity operator is easy to compute.

We will assume strong convexity of some functions: a convex function $\phi$ is said to be $\mu_\phi$-strongly convex, for some $\mu_\phi \geq 0$, if $\phi - \frac{\mu_\phi}{2}\|\cdot\|^2$ is convex. This covers the case $\mu_\phi = 0$, in which $\phi$ is merely convex.

### 2.1 PROXIMITY OPERATORS AND PROXIMAL ALGORITHMS

We recall that for any function $\phi$ and parameter $\gamma > 0$, the proximity operator of $\gamma\phi$ is (Bauschke & Combettes, 2017): $\text{prox}_{\gamma\phi} : x \in \mathcal{X} \mapsto \arg\min_{x' \in \mathcal{X}} \big( \gamma\phi(x') + \frac{1}{2}\|x' - x\|^2 \big)$. This operator has a closed form for many functions of practical interest (Parikh & Boyd, 2014; Pustelnik & Condat, 2017; Gheche et al., 2018), see also the website `http://proximity-operator.net`. In addition, the Moreau identity holds:

$$\text{prox}_{\gamma\phi^*}(x) = x - \gamma\,\text{prox}_{\phi/\gamma}(x/\gamma),$$

where $\phi^* : x \in \mathcal{X} \mapsto \sup_{x' \in \mathcal{X}} \big( \langle x, x' \rangle - \phi(x') \big)$ denotes the conjugate function of $\phi$ (Bauschke & Combettes, 2017). Thus, one can compute the proximity operator of $\phi$ from the one of $\phi^*$, and conversely.

Proximal splitting algorithms, such as the forward–backward and the Douglas–Rachford algorithms (Bauschke & Combettes, 2017), are well suited to minimizing the sum, $f + g$ or $g + h$ in our notation, of two functions. However, many problems take the form (1) with $K \neq \mathrm{Id}$, where $\mathrm{Id}$ denotes the identity, and the proximity operator of $h \circ K$ is intractable in most cases. A classical example is the total variation, widely used in image processing (Rudin et al., 1992; Caselles et al., 2011; Condat, 2014; 2017) or for regularization on graphs (Couprie et al., 2013), where $h$ is some variant of the $\ell_1$ norm and $K$ takes differences between adjacent values. Another example is when $h$ is the indicator function of some nonempty closed convex set $\Omega \subset \mathcal{U}$; that is, $h(u) = (0$ if $u \in \Omega$, $+\infty$ otherwise), in which case the problem (1) can be rewritten as

$$\text{Find } x^\star \in \arg\min_{x \in \mathcal{X}} \Big( f(x) + g(x) \Big) \quad \text{s.t.} \quad Kx \in \Omega.$$

If $g = 0$ and $\Omega = \{b\}$ for some $b \in \mathrm{ran}(K)$, where ran denotes the range, the problem can be further rewritten as the linearly constrained smooth minimization problem

$$\text{Find } x^\star \in \arg\min_{x \in \mathcal{X}} f(x) \quad \text{s.t.} \quad Kx = b.$$

This last problem has applications in decentralized optimization, for instance (Xin et al., 2020; Kovalev et al., 2020; Salim et al., 2022a). Thus, the template problem (1) covers a wide range of optimization problems met in machine learning (Bach et al., 2012; Polson et al., 2015), signal and image processing (Combettes & Pesquet, 2010; Chambolle & Pock, 2016), control (Stathopoulos et al., 2016), and many other fields. Examples include compressed sensing (Candès et al., 2006), object discovery in computer vision (Vo et al., 2019), $\ell_1$ trend filtering (Kim et al., 2009), group lasso (Yuan & Lin, 2006), square-root lasso (Belloni et al., 2011), Dantzig selector (Candès & Tao, 2007), and support-vector machines (Cortes & Vapnik, 1995).

## 2.2 THE DUAL PROBLEM, SADDLE-POINT REFORMULATION, AND OPTIMALITY CONDITIONS

In order to analyze algorithms solving such problems, we introduce the dual problem to (1):

$$\text{Find } u^\star \in \arg\min_{u \in \mathcal{U}} \Big( (f + g)^*(-K^*u) + h^*(u) \Big), \tag{2}$$

where $K^* : \mathcal{U} \to \mathcal{X}$ is the adjoint operator of $K$. We can also express the primal and dual problems as a combined saddle-point problem:

$$\text{Find } (x^\star, u^\star) \in \arg\min_{x \in \mathcal{X}} \max_{u \in \mathcal{U}} \Big( f(x) + g(x) + \langle Kx, u \rangle - h^*(u) \Big). \tag{3}$$

For these problems to be well-posed, we suppose that there exists $x^\star \in \mathcal{X}$ such that

$$0 \in \nabla f(x^\star) + \partial g(x^\star) + K^*\partial h(Kx^\star), \tag{4}$$

where $\partial(\cdot)$ denotes the subdifferential (Bauschke & Combettes, 2017). By Fermat's rule, every $x^\star$ satisfying (4) is a solution to (1). Equivalently to (4), we suppose that there exists $(x^\star, u^\star) \in \mathcal{X} \times \mathcal{U}$ such that

$$\begin{cases} 0 \in \nabla f(x^\star) + \partial g(x^\star) + K^*u^\star \\ 0 \in -Kx^\star + \partial h^*(u^\star) \end{cases}. \tag{5}$$

Every $(x^\star, u^\star)$ satisfying (5) is a primal–dual solution pair; that is, $x^\star$ is a solution to (1), $u^\star$ is a solution to (2), and $(x^\star, u^\star)$ is a solution to (3).

## 3 PROPOSED ALGORITHM: RandProx

There exist several deterministic algorithms for solving the problem (1); see Condat et al. (2023a) for a recent overview. In this work, we focus on the PDDY algorithm (Algorithm 1) (Salim et al., 2022b; Condat et al., 2022c). In particular, our new algorithm RandProx (Algorithm 2) generalizes the PDDY algorithm with a stochastic estimate of the proximity operator of $h^*$.

| **Algorithm 1** PDDY algorithm (Salim et al., 2022b) | **Algorithm 2** RandProx [new] |
|---|---|
| **input:** initial points $x^0 \in \mathcal{X}$, $u^0 \in \mathcal{U}$; 
 stepsizes $\gamma > 0$, $\tau > 0$ 
 $v^0 := K^* u^0$ 
 **for** $t = 0, 1, \ldots$ **do** 
 $\quad \hat{x}^t := \mathrm{prox}_{\gamma g}\big(x^t - \gamma \nabla f(x^t) - \gamma v^t\big)$ 
 $\quad u^{t+1} := \mathrm{prox}_{\tau h^*}\big(u^t + \tau K \hat{x}^t\big)$ 
 $\quad v^{t+1} := K^* u^{t+1}$ 
 $\quad x^{t+1} := \hat{x}^t - \gamma(v^{t+1} - v^t)$ 
 **end for** | **input:** initial points $x^0 \in \mathcal{X}$, $u^0 \in \mathcal{U}$; 
 stepsizes $\gamma > 0$, $\tau > 0$; $\omega \geq 0$ 
 $v^0 := K^* u^0$ 
 **for** $t = 0, 1, \ldots$ **do** 
 $\quad \hat{x}^t := \mathrm{prox}_{\gamma g}\big(x^t - \gamma \nabla f(x^t) - \gamma v^t\big)$ 
 $\quad u^{t+1} := u^t + \frac{1}{1+\omega}\mathcal{R}^t\big(\mathrm{prox}_{\tau h^*}(u^t + \tau K \hat{x}^t) - u^t\big)$ 
 $\quad v^{t+1} := K^* u^{t+1}$ 
 $\quad x^{t+1} := \hat{x}^t - \gamma\,(1+\omega)\,(v^{t+1} - v^t)$ 
 **end for** |

### 3.1 THE PDDY ALGORITHM

We recall the general convergence result for the PDDY algorithm (Condat et al., 2022c, Theorem 2):

> *If $\gamma \in (0, 2/L_f)$, $\tau > 0$, $\tau\gamma\|K\|^2 \leq 1$, then $(x^t)_{t\in\mathbb{N}}$ converges to a primal solution $x^\star$ of (1) and $(u^t)_{t\in\mathbb{N}}$ converges to a dual solution $u^\star$ of (2).*

The PDDY algorithm is similar and closely related to the PD3O algorithm (Yan, 2018), as discussed in Salim et al. (2022b); Condat et al. (2022c). It is also an instance (Algorithm 5) of the Asymmetric Forward–Backward Adjoint (AFBA) framework of Latafat & Patrinos (2017). We note that the popular Condat–Vũ algorithm (Condat, 2013; Vũ, 2013) can solve the same problem but has more restrictive conditions on $\gamma$ and $\tau$.

In the PDDY algorithm, the full gradient $\nabla f$ can be replaced by a stochastic estimator which is typically cheaper to compute (Salim et al., 2022b). Convergence rates and accelerations of the PDDY algorithm, as well as distributed versions of the algorithm, have been derived in Condat et al. (2022c). In particular, if $\mu_f > 0$ or $\mu_g > 0$, the primal problem (1) is strongly convex. In this case, a varying stepsize strategy accelerates the algorithm, with a $\mathcal{O}(1/t^2)$ decay of $\|x^t - x^\star\|^2$, where $x^\star$ is the unique solution to (1). But strong convexity of the primal problem is not sufficient for the PDDY algorithm to converge linearly, and additional assumptions on $h$ and $K$ are needed. We will prove linear convergence when both the primal and dual problems are strongly convex; this is a natural condition for primal–dual algorithms.

We note that $h$ is $L_h$-smooth, for some $L_h > 0$, if and only if $h^*$ is $\mu_{h^*}$-strongly convex, for some $\mu_{h^*} > 0$, with $\mu_{h^*} = 1/L_h$. In that case, the dual problem (2) is strongly convex.

### 3.2 RANDOMIZATION MECHANISM FOR THE PROXIMITY OPERATOR OF $h^*$

We propose RandProx (Algorithm 2), a generalization of the PDDY algorithm (Algorithm 1) with a randomized update of the dual variable $u$. Let us formalize the random operations using random variables and stochastic processes. We introduce the underlying probability space $(\mathcal{S}, \mathcal{F}, P)$. Given a real Hilbert space $\mathcal{H}$, an $\mathcal{H}$-valued random variable is a measurable map from $(\mathcal{S}, \mathcal{F})$ to $(\mathcal{H}, \mathcal{B})$, where $\mathcal{B}$ is the Borel $\sigma$-algebra of $\mathcal{H}$. Formally, randomizing some steps in the PDDY algorithm amounts to defining $\big((x^t, u^t)\big)_{t\in\mathbb{N}}$ as a stochastic process, with $x^t$ being a $\mathcal{X}$-valued random variable and $u^t$ a $\mathcal{U}$-valued random variable, for every $t \geq 0$. We use light notations and write our randomized algorithm RandProx using stochastic operators $\mathcal{R}^t$ on $\mathcal{U}$; that is, for every $t \geq 0$ and any $r^t \in \mathcal{U}$, $\mathcal{R}^t(r^t)$ is a $\mathcal{U}$-valued random variable, which can be interpreted as $r^t$ plus 'random noise' (formally, $r^t$ is itself a $\mathcal{U}$-valued random variable, but algorithmically, $\mathcal{R}^t$ is applied to a particular outcome in $\mathcal{U}$, hence the notation as an operator on $\mathcal{U}$). To fix the ideas, let us give two examples.

**Example 1.** The first example is compression (Alistarh et al., 2017; 2018; Horváth et al., 2022; Mishchenko et al., 2019; Albasyoni et al., 2020; Beznosikov et al., 2020; Condat et al., 2022b): $\mathcal{U} = \mathbb{R}^d$ for some $d \geq 1$ and $\mathcal{R}^t$ is the well known rand-$k$ compressor or sparsifier, with $1 \leq k < d$: $\mathcal{R}^t$ multiplies $k$ coordinates, chosen uniformly at random, of the vector $r^t$ by $d/k$ and sets the other ones to zero. An application to compressed communication is discussed in Section A.3.

**Example 2.** The second example, discussed in Section A.1, is the Bernoulli, or coin flip, operator

$$\mathcal{R}^t : r^t \mapsto \begin{cases} \frac{1}{p} r^t & \text{with probability } p, \\ 0 & \text{with probability } 1 - p, \end{cases} \tag{6}$$

for some $p > 0$. In that case, with probability $1 - p$, the outcome of $\mathcal{R}^t(r^t)$ is 0 and $r^t$ does not need to be calculated; in particular, in the RandProx algorithm, $\text{prox}_{\tau h^*}$ is not called, and this is why one can expect the iteration complexity of RandProx to decrease. Thus, in this example, $\mathcal{R}^t(r^t)$ does not really consist of applying the operator $\mathcal{R}^t$ to $r^t$; in general, the notation $\mathcal{R}^t(r^t)$ simply denotes a stochastic estimate of $r^t$.

**Example 3.** The third example, discussed in Section A.2, is sampling, which makes it possible to solve problems involving a sum $\sum_{i=1}^{n} h_i$ of functions, by calling the proximity operator of only one randomly chosen function $h_i$, instead of all functions, at every iteration. The Point-SAGA algorithm (Defazio, 2016) is recovered as a particular case of RandProx in this setting.

Hereafter, we denote by $\mathcal{F}_t$ the $\sigma$-algebra generated by the collection of $(\mathcal{X} \times \mathcal{U})$-valued random variables $(x^0, u^0), \ldots, (x^t, u^t)$, for every $t \geq 0$. In this work, we consider **unbiased** random estimates: for every $t \geq 0$,

$$\mathbb{E}\big[\mathcal{R}^t(r^t) \mid \mathcal{F}_t\big] = r^t,$$

where $\mathbb{E}[\cdot]$ denotes the expectation, here conditionally on $\mathcal{F}_t$, and $r^t$ is the random variable

$$r^t := \text{prox}_{\tau h^*}(u^t + \tau K \hat{x}^t) - u^t,$$

as defined by RandProx. Note that our framework is general in that for $t \neq t'$, $\mathcal{R}^t$ and $\mathcal{R}^{t'}$ need not be independent nor have the same law. In simple words, at every iteration, the randomness is new but can have a different form and depend on the past, so that the operators $\mathcal{R}^t$ can be defined dynamically on the fly in RandProx.

We characterize the operators $\mathcal{R}^t$ by their *relative variance* $\omega \geq 0$ such that, for every $t \geq 0$,

$$\mathbb{E}\Big[\big\|\mathcal{R}^t(r^t) - r^t\big\|^2 \mid \mathcal{F}_t\Big] \leq \omega \big\|r^t\big\|^2. \tag{7}$$

This assumption is satisfied by a large class of randomization strategies, which are widely used to define unbiased stochastic gradient estimates. We refer to Beznosikov et al. (2020), Table 1 in Safaryan et al. (2021), Zhang et al. (2023), Szlendak et al. (2022) for examples. In the Example 1 above of `rand-k`, $\omega = \frac{d}{k} - 1$. In Example 2, $\omega = \frac{1}{p} - 1$. In Example 3, $\omega = n - 1$. The value of $\omega$ is supposed known and is used in the RandProx algorithm. Note that $\omega = 0$ if and only if $\mathcal{R}^t = \text{Id}$, in which case there is no randomness and RandProx reverts to the original deterministic PDDY algorithm.

Thus, $\mathcal{R}^t(r^t) = r^t + e^t$, with the variance of the error $e^t$ proportional to $\|r^t\|^2$. In particular, if $r^t = 0$, there is no error and $\mathcal{R}^t(0) = 0$. The stochastic operators $\mathcal{R}^t$ will be applied to a sequence of random vectors that will converge to zero, and hence the error will converge to zero as well, due to the relative variance property (7). RandProx is therefore a *variance-reduced* method (Hanzely & Richtárik, 2019; Gorbunov et al., 2020; Gower et al., 2020): the random errors vanish along the iterations and the algorithm converges to an exact solution of the problem.

To characterize how the error on the dual variable propagates to the primal variable after applying $K^*$, we also introduce the relative variance $\omega_{\text{ran}} \geq 0$ in the range of $K^*$ and the offset $\zeta \in [0, 1]$ such that, for every $t \geq 0$,

$$\mathbb{E}\Big[\big\|K^*\big(\mathcal{R}^t(r^t) - r^t\big)\big\|^2 \mid \mathcal{F}_t\Big] \leq \omega_{\text{ran}} \big\|r^t\big\|^2 - \zeta \big\|K^* r^t\big\|^2. \tag{8}$$

It is easy to see that (8) holds with $\omega_{\text{ran}} = \|K\|^2 \omega$ and $\zeta = 0$, so this is the default choice without particular knowledge on $K^*$. But in some situations, e.g. sampling like in Section A.2, a much smaller value of $\omega_{\text{ran}}$ and a positive value of $\zeta$ can be derived.

### 3.3 DESCRIPTION OF THE ALGORITHM

Let us now describe how the PDDY and RandProx algorithms work. An iteration consists in 3 steps:

1. Given $x^t$ and $u^t$, the updated value of the primal variable is *predicted* to be $\hat{x}^t$.

2. The points $\hat{x}^t$ and $u^t$ are used to update the dual variable to its new value $u^{t+1}$.

3. The primal variable is *corrected* from $\hat{x}^t$ to $x^{t+1}$, by back-propagating the difference $u^{t+1} - u^t$ using $K^*$.

In RandProx, randomization takes place in Step 2. On average, this decreases the progress from $u^t$ to $u^{t+1}$, and in turn from $\hat{x}^t$ to $x^{t+1}$ in Step 3, but the progress from $x^t$ to $\hat{x}^t$, due to the unaltered proximal gradient descent step in Step 1, is kept. Therefore, randomization can be used to balance the progress speed on the primal and dual variables, depending on the relative computational complexity of the gradient and proximity operators. The random errors are kept under control and convergence is ensured using *underrelaxation*: let us define, for every $t \geq 0$,

$$\hat{u}^{t+1} := \operatorname{prox}_{\tau h^*}\left(u^t + \tau K \hat{x}^t\right). \tag{9}$$

The PDDY algorithm updates the dual variable by setting $u^{t+1} := \hat{u}^{t+1}$. In RandProx, let us define

$$\tilde{u}^{t+1} := u^t + \mathcal{R}^t\left(\hat{u}^{t+1} - u^t\right) = \hat{u}^{t+1} + e^t$$

for some zero-mean random error $e^t$, keeping in mind that $\tilde{u}^{t+1}$ is typically cheaper to compute than $\hat{u}^{t+1}$. Then underrelaxation is applied: we set

$$u^{t+1} := \rho \tilde{u}^{t+1} + (1 - \rho) u^t \tag{10}$$

for some relaxation parameter $\rho \in (0, 1]$; we use $\rho = \frac{1}{1+\omega}$ in the algorithm. That is, the update of the dual variable consists in a convex combination of the old estimate $u^t$ and the new, better in expectation but noisy, estimate $\tilde{u}^{t+1}$. Noise is mitigated by underrelaxation, because the error $e^t$ is multiplied by $\rho$, so that its variance is multiplied by $\rho^2$. So, even if $\omega$ is arbitrarily large, $\omega \rho^2$ is kept small. Underrelaxation slows down the progress on the dual variable of the algorithm towards the solution, but if the iterations become faster, this is beneficial overall.

## 4 CONVERGENCE ANALYSIS OF RandProx

Our most general result, whose proof is in the Appendix, is the following:

**Theorem 1.** *Suppose that $\mu_f > 0$ or $\mu_g > 0$, and that $\mu_{h^*} > 0$. In RandProx, suppose that $0 < \gamma < \frac{2}{L_f}$, $\tau > 0$, and $\gamma \tau \left((1 - \zeta)\|K\|^2 + \omega_{\mathrm{ran}}\right) \leq 1$, where $\omega_{\mathrm{ran}}$ and $\zeta$ are defined in (8).[1] For every $t \geq 0$, define the Lyapunov function*

$$\Psi^t := \frac{1}{\gamma}\left\|x^t - x^\star\right\|^2 + (1 + \omega)\left(\frac{1}{\tau} + 2\mu_{h^*}\right)\left\|u^t - u^\star\right\|^2, \tag{11}$$

*where $x^\star$ and $u^\star$ are the unique solutions to (1) and (2), respectively. Then RandProx converges linearly: for every $t \geq 0$,*

$$\mathbb{E}\left[\Psi^t\right] \leq c^t \Psi^0, \tag{12}$$

*where*

$$c := \max\left(\frac{(1 - \gamma\mu_f)^2}{1 + \gamma\mu_g}, \frac{(\gamma L_f - 1)^2}{1 + \gamma\mu_g}, 1 - \frac{2\tau\mu_{h^*}}{(1 + \omega)(1 + 2\tau\mu_{h^*})}\right) < 1. \tag{13}$$

*Also, $(x^t)_{t\in\mathbb{N}}$ and $(\hat{x}^t)_{t\in\mathbb{N}}$ both converge to $x^\star$ and $(u^t)_{t\in\mathbb{N}}$ converges to $u^\star$, almost surely.*

In Theorem 1, if $\gamma \leq \frac{2}{L_f + \mu_f}$, we have $\max(1 - \gamma\mu_f, \gamma L_f - 1)^2 = (1 - \gamma\mu_f)^2 \leq 1 - \gamma\mu_f$, so that in that case the rate $c$ in (13) satisfies

$$c \leq 1 - \min\left(\frac{\gamma(\mu_f + \mu_g)}{1 + \gamma\mu_g}, \frac{2\tau\mu_{h^*}}{(1 + \omega)(1 + 2\tau\mu_{h^*})}\right) < 1.$$

---

[1]The condition $\gamma < \frac{2}{L_f}$ is given for simplicity. Larger values of $\gamma$ can be used when $\mu_g > 0$, as long as $c < 1$ in (13).

Table 1: The different particular cases of the problem (1) for which we derive an instance of RandProx, with the number of the theorem where its linear convergence is stated, and the corresponding condition on $h$ and $K$. $\lambda$ is a shorthand notation for $\lambda_{\min}(KK^*)$ and $\imath_{\{b\}} : x \mapsto (0$ if $x = b, +\infty$ otherwise$)$.

| $f$ | $g$ | $h$ | $K$ | Deterministic algorithm | Randomized algorithm | Theorem | Condition ensuring linear convergence |
|-----|-----|-----|-----|------------------------|----------------------|---------|--------------------------------------|
| any | any | any | any | PDDY | RandProx | 1 | $\mu_{h^*} > 0$ |
| any | 0 | any | any | PAPC | RandProx | 2 | $\mu_{h^*} > 0$ or $\lambda > 0$ |
| any | 0 | any | Id | forward-backward (FB) | RandProx-FB | 3 | — |
| any | 0 | $\imath_{\{b\}}$ | any | PAPC | RandProx-LC | 4 | — |
| 0 | any | any | any | Chambolle–Pock (CP) | RandProx-CP | 7 | $\mu_{h^*} > 0$ |
| 0 | any | any | Id | ADMM | RandProx-ADMM | 8 | $\mu_{h^*} > 0$ |
| any | any | any | Id | Davis–Yin (DY) | RandProx-DY | 9 | $\mu_{h^*} > 0$ |

**Remark 1** (choice of $\tau$) Given $\gamma$, the rate $c$ in (13) is smallest if $\tau$ is largest. So, there seems to be no reason to take $\tau\gamma\big((1 - \zeta)\|K\|^2 + \omega_{\mathrm{ran}}\big) < 1$, and $\tau\gamma\big((1 - \zeta)\|K\|^2 + \omega_{\mathrm{ran}}\big) = 1$ should be the best choice in most cases. Thus, one can set $\tau = \frac{1}{\gamma((1-\zeta)\|K\|^2+\omega_{\mathrm{ran}})}$ and keep $\gamma$ as the only parameter to tune in RandProx.

In the rest of this section, we discuss some particular cases of (1), for which we derive stronger convergence guarantees than in Theorem 1 for RandProx. Other particular cases are studied in the Appendix; for instance, an instance of RandProx, called RandProx-ADMM, is a randomized version of the popular ADMM (Boyd et al., 2011). The different particular cases are summarized in Table 1.

## 4.1 PARTICULAR CASE $g = 0$

In this section, we assume that $g = 0$. Then the PDDY algorithm becomes an algorithm proposed for least-squares problems (Loris & Verhoeven, 2011) and rediscovered independently as the PDFP2O algorithm (Chen et al., 2013) and as the Proximal Alternating Predictor-Corrector (PAPC) algorithm (Drori et al., 2015); let us call it the PAPC algorithm. It has been shown to have a primal–dual forward–backward structure (Combettes et al., 2014). Thus, when $g = 0$, RandProx is a randomized version of the PAPC algorithm.

We note that $f^*$ is strongly convex, which is not the case of $(f + g)^*$ in general. Let us define $\lambda_{\min}(KK^*)$ as the smallest eigenvalue of $KK^*$. $\lambda_{\min}(KK^*) > 0$ if and only if $\ker(K^*) = \{0\}$, where ker denotes the kernel. If $\lambda_{\min}(KK^*) > 0$, $f^*(-K^*\cdot)$ is strongly convex. Thus, when $g = 0$, $\lambda_{\min}(KK^*) > 0$ and $\mu_{h^*} > 0$ are two sufficient conditions for the dual problem (2) to be strongly convex. We indeed get linear convergence of RandProx in that case:

**Theorem 2.** *Suppose that* $g = 0$, $\mu_f > 0$, *and that* $\lambda_{\min}(KK^*) > 0$ *or* $\mu_{h^*} > 0$. *In* RandProx, *suppose that* $0 < \gamma < \frac{2}{L_f}$, $\tau > 0$ *and* $\gamma\tau\big((1 - \zeta)\|K\|^2 + \omega_{\mathrm{ran}}\big) \leq 1$. *Then* RandProx *converges linearly: for every* $t \geq 0$, $\mathbb{E}[\Psi^t] \leq c^t\Psi^0$, *where the Lyapunov function* $\Psi^t$ *is defined in* (11), *and*

$$c := \max\left((1 - \gamma\mu_f)^2, (\gamma L_f - 1)^2, 1 - \frac{2\tau\mu_{h^*} + \gamma\tau\lambda_{\min}(KK^*)}{(1 + \omega)(1 + 2\tau\mu_{h^*})}\right) < 1. \qquad (14)$$

*Also,* $(x^t)_{t\in\mathbb{N}}$ *and* $(\hat{x}^t)_{t\in\mathbb{N}}$ *both converge to* $x^\star$ *and* $(u^t)_{t\in\mathbb{N}}$ *converges to* $u^\star$, *almost surely.*

When $\mathcal{R}^t = \mathrm{Id}$ and $\omega = \omega_{\mathrm{ran}} = 0$, RandProx reverts to the PAPC algorithm. Even in this particular case, Theorem 2 proves linear convergence of the PAPC algorithm and is new. In Chen et al. (2013, Theorem 3.7), the authors proved linear convergence of an underrelaxed version of the algorithm; underrelaxation slows down convergence. In Luke & Shefi (2018), Theorem 3.1 is wrong, since it is based on the false assumption that if $\lambda_{\min}(K_iK_i^*) > 0$ for linear operators $K_i$, $i = 1, \ldots, p$, then $\lambda_{\min}(KK^*) > 0$, with $K : x \mapsto (K_1x, \ldots, K_px)$. Their theorem remains valid when $p = 1$, but their rate is complicated and worse than ours.

We now consider the even more particular case of $g = 0$ and $K = \mathrm{Id}$. Then the problems (1) and (2) consist in minimizing $f(x) + h(x)$ and $f^*(-u) + h^*(u)$, respectively. The dual problem is strongly convex and has a unique solution $u^\star = -\nabla f(x^\star)$, for any primal solution $x^\star$. By setting $\tau := 1/\gamma$

---

**Algorithm 3** RandProx-FB [new]

---

  **input:** initial points $x^0 \in \mathcal{X}$, $u^0 \in \mathcal{X}$;
  stepsize $\gamma > 0$; $\omega \geq 0$
  **for** $t = 0, 1, \ldots$ **do**
    $\hat{x}^t := x^t - \gamma \nabla f(x^t) - \gamma u^t$
    $d^t := \mathcal{R}^t \big( \hat{x}^t - \mathrm{prox}_{\gamma(1+\omega)h}(\hat{x}^t + \gamma(1+\omega)u^t) \big)$
    $u^{t+1} := u^t + \frac{1}{\gamma(1+\omega)^2} d^t$
    $x^{t+1} := \hat{x}^t - \frac{1}{1+\omega} d^t$
  **end for**

---

**Algorithm 4** RandProx-LC [new]

---

  **input:** initial points $x^0 \in \mathcal{X}$, $u^0 \in \mathcal{U}$;
  stepsizes $\gamma > 0$, $\tau > 0$; $\omega \geq 0$
  $v^0 := K^* u^0$
  **for** $t = 0, 1, \ldots$ **do**
    $\hat{x}^t := x^t - \gamma \nabla f(x^t) - \gamma v^t$
    $u^{t+1} := u^t + \frac{\tau}{1+\omega} \mathcal{R}^t (K\hat{x}^t - b)$
    $v^{t+1} := K^* u^{t+1}$
    $x^{t+1} := \hat{x}^t - \gamma(1+\omega)(v^{t+1} - v^t)$
  **end for**

---

in the PAPC algorithm, we obtain the classical proximal gradient, a.k.a. forward-backward (FB), algorithm, which iterates $x^{t+1} := \mathrm{prox}_{\gamma h} \big( x^t - \gamma \nabla f(x^t) \big)$. Thus, when randomness is introduced, we set $\omega_{\mathrm{ran}} := \omega$, $\zeta := 0$ and, according to Remark 1, $\tau := \frac{1}{\gamma(1+\omega)}$ in RandProx. By noting that, for every $a > 0$, the abstract operators $\mathcal{R}^t$ and $a\mathcal{R}^t \big( \frac{1}{a} \cdot \big)$ have the same properties, we can put the constant $\gamma(1 + \omega)$ outside $\mathcal{R}^t$ to simplify the algorithm, and rewrite RandProx as RandProx-FB, shown above. As a corollary of Theorem 2, we have:

**Theorem 3.** *Suppose that $\mu_f > 0$. In RandProx-FB, suppose that $0 < \gamma < \frac{2}{L_f}$. For every $t \geq 0$, define the Lyapunov function*

$$\Psi^t := \frac{1}{\gamma} \left\| x^t - x^\star \right\|^2 + (1 + \omega)\big(\gamma(1+\omega) + 2\mu_{h^*}\big) \left\| u^t - u^\star \right\|^2, \tag{15}$$

*where $x^\star$ is the unique minimizer of $f + h$ and $u^\star = -\nabla f(x^\star)$ is the unique minimizer of $f^*(-\cdot) + h^*$. Then RandProx-FB converges linearly: for every $t \geq 0$,*

$$\mathbb{E}\big[\Psi^t\big] \leq c^t \Psi^0,$$

*where*

$$c := \max \left( (1 - \gamma\mu_f)^2, (\gamma L_f - 1)^2, 1 - \frac{1 + \frac{2}{\gamma}\mu_{h^*}}{(1 + \omega)\big(1 + \omega + \frac{2}{\gamma}\mu_{h^*}\big)} \right) < 1. \tag{16}$$

*Also, $(x^t)_{t \in \mathbb{N}}$ and $(\hat{x}^t)_{t \in \mathbb{N}}$ both converge to $x^\star$ and $(u^t)_{t \in \mathbb{N}}$ converges to $u^\star$, almost surely.*

It is important to note that it is not necessary to have $\mu_{h^*} > 0$ in Theorem 3. If we ignore the properties of $h^*$, the third factor in (16) can be replaced by its upper bound $1 - \frac{1}{(1+\omega)^2}$.

## 4.2 LINEARLY CONSTRAINED SMOOTH MINIMIZATION

Let $b \in \mathrm{ran}(K)$. In this section, we consider the linearly constrained (LC) minimization problem

$$\text{Find } x^\star \in \arg\min_{x \in \mathcal{X}} f(x) \quad \text{s.t.} \quad Kx = b, \tag{17}$$

which is a particular case of (1) with $g = 0$ and $h : u \in \mathcal{U} \mapsto (0 \text{ if } u = b, +\infty \text{ otherwise})$. We have $h^* : u \in \mathcal{U} \mapsto \langle u, b \rangle$ and $\mathrm{prox}_{\tau h^*} : u \in \mathcal{U} \mapsto u - \tau b$. The dual problem to (17) is

$$\text{Find } u^\star \in \arg\min_{u \in \mathcal{U}} \left( f^*(-K^*u) + \langle u, b \rangle \right). \tag{18}$$

We denote by $u_0^\star$ the unique solution to (18) in $\mathrm{ran}(K)$. Then the set of solutions of (18) is the affine subspace $u_0^\star + \ker(K^*)$. Thus, the dual problem is not strongly convex, unless $\ker(K^*) = \{0\}$. Yet, we will see that strong convexity of $f$ is sufficient to have linear convergence of RandProx, without any condition on $K$.

We rewrite RandProx in this setting as RandProx-LC, shown above. We observe that $u^t$ does not appear in the argument of $\mathcal{R}^t$ any more, so that the iteration can be rewritten with the variable $v^t = K^* u^t$, and $u^t$ can be removed if we are not interested in estimating a dual solution. In any case,

we denote by $P_{\mathrm{ran}(K)}$ the orthogonal projector onto $\mathrm{ran}(K)$ and by $\lambda_{\min}^+(KK^*) > 0$ the smallest *nonzero* eigenvalue of $KK^*$. Then:

**Theorem 4.** *In the setup* (17)–(18), *suppose that* $\mu_f > 0$. *In* RandProx-LC, *suppose that* $0 < \gamma < \frac{2}{L_f}$, $\tau > 0$ *and* $\gamma\tau\big((1-\zeta)\|K\|^2 + \omega_{\mathrm{ran}}\big) \leq 1$. *Define the Lyapunov function, for every* $t \geq 0$,

$$\Psi^t := \frac{1}{\gamma}\left\|x^t - x^\star\right\|^2 + \frac{1+\omega}{\tau}\left\|u_0^t - u_0^\star\right\|^2, \tag{19}$$

*where* $u_0^t := P_{\mathrm{ran}(K)}(u^t)$ *is also the unique element in* $\mathrm{ran}(K)$ *such that* $v^t = K^* u_0^t$, $x^\star$ *is the unique solution of* (17) *and* $u_0^\star$ *is the unique solution in* $\mathrm{ran}(K)$ *of* (18). *Then* RandProx-LC *converges linearly: for every* $t \geq 0$,

$$\mathbb{E}\big[\Psi^t\big] \leq c^t\Psi^0,$$

*where*

$$c := \max\left((1 - \gamma\mu_f)^2, (\gamma L_f - 1)^2, 1 - \frac{\gamma\tau\lambda_{\min}^+(KK^*)}{1+\omega}\right) < 1. \tag{20}$$

*Also,* $(x^t)_{t\in\mathbb{N}}$ *and* $(\hat{x}^t)_{t\in\mathbb{N}}$ *both converge to* $x^\star$ *and* $(u_0^t)_{t\in\mathbb{N}}$ *converges to* $u_0^\star$, *almost surely.*

Theorem 4 is new even for the PAPC algorithm when $\omega = 0$: its linear convergence under the stronger condition $\gamma\tau\|K\|^2 < 1$ has been shown in Salim et al. (2022b, Theorem 6.2), but our rate in (20) is better.

We further discuss RandProx-LC, which can be used for decentralized optimization, in the Appendix. Another example of application is when $\mathcal{X} = \mathbb{R}^d$, for some $d \geq 1$, and $K$ is a matrix; one can solve (17) by activating one row of $K$ chosen uniformly at random at every iteration.

## 5 CONVERGENCE IN THE MERELY CONVEX CASE

In all theorems, strong convexity of $f$ or $g$ is assumed; that is, $\mu_f > 0$ or $\mu_g > 0$. In this section, we remove this hypothesis, so that the primal problem is not necessarily strongly convex any more. But $\nabla f(x^\star)$ is the same for every solution $x^\star$ of (1), and we denote by $\nabla f(x^\star)$ this element.

We define the Bregman divergence of $f$ at points $(x, x') \in \mathcal{X}^2$ as

$$D_f(x, x') := f(x) - f(x') - \langle\nabla f(x'), x - x'\rangle \geq 0.$$

For every $t \geq 0$, $D_f(x^t, x^\star)$ is the same for every solution $x^\star$ of (1), and we denote by $D_f(x^t, x^\star)$ this element. $D_f(x^t, x^\star)$ can be viewed as a generalization of the objective gap $f(x^t) - f(x^\star)$ to the case when $\nabla f(x^\star) \neq 0$. $D_f(x^t, x^\star)$ is a loose kind of distance between $x^t$ and the solution set, but under some additional assumptions on $f$, for instance strict convexity, $D_f(x^t, x^\star) \to 0$ implies that the distance from $x^t$ to the solution set tends to zero. Also, $D_f(x^t, x^\star) \geq \frac{1}{2L_f}\|\nabla f(x^t) - \nabla f(x^\star)\|^2$, so that $D_f(x^t, x^\star) \to 0$ implies that $\big(\nabla f(x^t)\big)_{t\in\mathbb{N}}$ converges to $\nabla f(x^\star)$.

**Theorem 11.** *In* RandProx, *suppose that* $0 < \gamma < \frac{2}{L_f}$, $\tau > 0$, *and* $\gamma\tau\big((1-\zeta)\|K\|^2 + \omega_{\mathrm{ran}}\big) \leq 1$. *Then* $D_f(x^t, x^\star) \to 0$, *almost surely and in quadratic mean. Moreover, for every* $t \geq 0$, *we define* $\bar{x}^t := \frac{1}{t+1}\sum_{i=0}^t x^i$. *Then, for every* $t \geq 0$,

$$\mathbb{E}\big[D_f(\bar{x}^t, x^\star)\big] \leq \frac{\Psi^0}{(2 - \gamma L_f)(t+1)} = \mathcal{O}(1/t). \tag{21}$$

*If, in addition,* $\mu_{h^*} > 0$, *there is a unique dual solution* $u^\star$ *to* (2) *and* $(u^t)_{t\in\mathbb{N}}$ *converges to* $u^\star$, *in quadratic mean.*

We can derive counterparts of the other theorems in the same way. These theorems apply to all algorithms presented in the paper. For instance, Theorem 11 applies to Scaffnew (Mishchenko et al., 2022), a particular case of RandProx-FL seen in Section A.3, and provides for it the first convergence results in the non-strongly convex case.

ACKNOWLEDGMENTS

The work of P. Richtárik was partially supported by the KAUST Baseline Research Fund Scheme and by the SDAIA-KAUST Center of Excellence in Data Science and Artificial Intelligence.

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

**Algorithm 5** RandProx-Skip [new]

> **input:** initial points $x^0 \in \mathcal{X}$, $u^0 \in \mathcal{U}$;
> stepsizes $\gamma > 0$, $\tau > 0$; $p \in (0, 1]$
> $v^0 := K^* u^0$
> **for** $t = 0, 1, \dots$ **do**
> $\quad \hat{x}^t := \mathrm{prox}_{\gamma g}\big(x^t - \gamma \nabla f(x^t) - \gamma v^t\big)$
> $\quad$ Flip a coin $\theta^t = (1$ with probability $p$,
> $\quad 0$ else$)$
> $\quad$ **if** $\theta^t = 1$ **then**
> $\quad\quad u^{t+1} := \mathrm{prox}_{\tau h^*}(u^t + \tau K \hat{x}^t)$
> $\quad\quad v^{t+1} := K^* u^{t+1}$
> $\quad\quad x^{t+1} := \hat{x}^t - \frac{\gamma}{p}(v^{t+1} - v^t)$
> $\quad$ **else**
> $\quad\quad u^{t+1} := u^t, v^{t+1} := v^t, x^{t+1} := \hat{x}^t$
> $\quad$ **end if**
> **end for**

**Algorithm 6** RandProx-Minibatch [new]

> **input:** initial points $x^0 \in \mathcal{X}$, $(u_i^0)_{i=1}^n \in \mathcal{X}^n$;
> stepsize $\gamma > 0$; $k \in \{1, \dots, n\}$
> $v^0 := \sum_{i=1}^n u_i^0$
> **for** $t = 0, 1, \dots$ **do**
> $\quad \hat{x}^t := \mathrm{prox}_{\gamma g}\big(x^t - \gamma \nabla f(x^t) - \gamma v^t\big)$
> $\quad$ pick $\Omega^t \subset \{1, \dots, n\}$ of size $k$ unif. at random
> $\quad$ **for** $i \in \Omega^t$ **do**
> $\quad\quad u_i^{t+1} := \mathrm{prox}_{\frac{1}{\gamma n} h_i^*}(u_i^t + \frac{1}{\gamma n} \hat{x}^t)$
> $\quad$ **end for**
> $\quad$ **for** $i \in \{1, \dots, n\} \backslash \Omega^t$ **do**
> $\quad\quad u_i^{t+1} := u_i^t$
> $\quad$ **end for**
> $\quad v^{t+1} := \sum_{i=1}^n u_i^{t+1}$
> $\quad x^{t+1} := \hat{x}^t - \frac{\gamma n}{k}(v^{t+1} - v^t)$
> **end for**

# Appendix

## A  EXAMPLES

### A.1  SKIPPING THE PROXIMITY OPERATOR

In this section, we consider the case of Bernoulli operators $\mathcal{R}^t$ defined in (6), which compute and return their argument only with probability $p > 0$. RandProx becomes RandProx-Skip, shown above. Then $\omega = \frac{1}{p} - 1$, $\omega_{\mathrm{ran}} = \|K\|^2 \omega$, and $\zeta = 0$.

If $g = 0$, RandProx-Skip reverts to the SplitSkip algorithm proposed recently (Mishchenko et al., 2022). Our Theorems 1 and 4 recover the same rate as given for SplitSkip in Mishchenko et al. (2022, Theorem D.1), if smoothness of $h$ is ignored. If in addition $K = \mathrm{Id}$ and $\tau = \frac{1}{\gamma(1+\omega)} = \frac{p}{\gamma}$, RandProx-Skip reverts to ProxSkip, a particular case of SplitSkip (Mishchenko et al., 2022). Our Theorem 3 applies to this case and allows us to exploit the possible smoothness of $h$ in RandProx-Skip = ProxSkip, which is not the case of the results of (Mishchenko et al., 2022). As a practical application of our new results, let us consider *personalized federated learning (FL)* (Hanzely et al., 2020): given a client-server architecture with a master and $n \geq 1$ users, each with local cost function $f_i$, $i = 1, \dots, n$, the goal is to

$$\underset{(x_i)_{i=1}^n \in (\mathbb{R}^d)^n}{\mathrm{minimize}} \ \sum_{i=1}^n f_i(x_i) + \frac{\lambda}{2} \sum_{i=1}^n \|x_i - \bar{x}\|^2, \tag{22}$$

where $\bar{x} := \frac{1}{n} \sum_{i=1}^n x_i$. Each $f_i$ is supposed $L_f$-smooth and $\mu_f$-strongly convex. We set $\mathcal{X} := (\mathbb{R}^d)^n$, $f : x = (x_i)_{i=1}^n \mapsto \sum_{i=1}^n f_i(x_i)$, $h : x \mapsto \frac{\lambda}{2} \sum_{i=1}^n \|x_i - \bar{x}\|^2$. $f$ is $L_f$-smooth and $\mu_f$-strongly convex, $h$ is $\lambda$-smooth, so that $\mu_{h^*} = \frac{1}{\lambda}$. Thus, with $\gamma = \frac{1}{L_f}$, we have in (16):

$$c \leq 1 - \min\left(\frac{\mu_f}{L_f}, \frac{1 + \frac{2L_f}{\lambda}}{\frac{1}{p}\left(\frac{1}{p} + \frac{2L_f}{\lambda}\right)}\right) < 1.$$

Hence, with $p = \frac{\sqrt{\mu_f \min(L_f, \lambda)}}{L_f} = \sqrt{\frac{\mu_f}{L_f}} \min\left(\sqrt{\frac{\lambda}{L_f}}, 1\right)$, the communication complexity in terms of the expected number of communication rounds to reach $\epsilon$-accuracy is $\mathcal{O}\left(\left(\sqrt{\frac{\min(L_f, \lambda)}{\mu_f}} + 1\right) \log \frac{1}{\epsilon}\right)$, which, up to the '+1' log factor, is optimal (Hanzely et al., 2020). This shows that in personalized FL with $\lambda < L_f$, the complexity can be decreased in comparison with non-personalized FL, which corresponds to $\lambda = +\infty$. This is achieved by properly setting $p$ in ProxSkip, according to our new theory, which exploits the smoothness of $h$.

### A.2 SAMPLING AMONG SEVERAL FUNCTIONS

We first remark that we can extend Problem (1) with the term $h(Kx)$ replaced by the sum $\sum_{i=1}^{n} h_i(K_i x)$ of $n \geq 2$ proper closed convex functions $h_i$ composed with linear operators $K_i : \mathcal{X} \to \mathcal{U}_i$, for some real Hilbert spaces $\mathcal{U}_i$, by using the classical product-space trick: by defining $\mathcal{U} := \mathcal{U}_1 \times \cdots \mathcal{U}_n$, $h : u = (u_i)_{i=1}^{n} \in \mathcal{U} \mapsto \sum_{i=1}^{n} h_i(u_i)$, $K : x \in \mathcal{X} \mapsto (K_i x)_{i=1}^{n} \in \mathcal{U}$, we have $h(Kx) = \sum_{i=1}^{n} h_i(K_i x)$. In particular, by setting $K_i := \mathrm{Id}$ and $\mathcal{U}_i := \mathcal{X}$, we consider in this section the problem:

$$\text{Find } x^\star \in \underset{x \in \mathcal{X}}{\arg\min} \left( f(x) + g(x) + \sum_{i=1}^{n} h_i(x) \right). \tag{23}$$

We have $h^* : (u_i)_{i=1}^{n} \in \mathcal{X}^n \mapsto \sum_{i=1}^{n} h_i^*(u_i)$ and we suppose that every function $h_i^*$ is $\mu_{h^*}$-strongly convex, for some $\mu_{h^*} \geq 0$; then $h^*$ is $\mu_{h^*}$-strongly convex. Thus, the dual problem to (23) is

$$\text{Find } (u_i^\star)_{i=1}^{n} \in \underset{(u_i)_{i=1}^{n} \in \mathcal{X}^n}{\arg\min} \left( (f+g)^* \left( -\sum_{i=1}^{n} u_i \right) + \sum_{i=1}^{n} h_i^*(u_i) \right). \tag{24}$$

Since $K^*K = n\mathrm{Id}$, $\|K\|^2 = n$. Now, we choose $\mathcal{R}^t$ as the `rand-k` sampling operator, for some $k \in \{1, \ldots, n\}$: $\mathcal{R}^t$ multiplies $k$ elements out of the $n$ of its argument sequence, chosen uniformly at random, by $n/k$ and sets the other ones to zero. It is known (Condat & Richtárik, 2022, Proposition 1) that we can set

$$\omega := \frac{n}{k} - 1, \quad \omega_{\mathrm{ran}} := \frac{n(n-k)}{k(n-1)}, \quad \zeta := \frac{n-k}{k(n-1)}.$$

Note that this value of $\omega_{\mathrm{ran}}$ is $n-1$ times smaller than the naive bound $\|K\|^2 \omega = \frac{n(n-k)}{k}$. We have $(1-\zeta)\|K\|^2 + \omega_{\mathrm{ran}} = n$. RandProx in this setting, with $\tau := \frac{1}{\gamma n}$, becomes RandProx-Minibatch, shown above, and Theorem 1 yields:

**Theorem 5.** *Suppose that $\mu_f > 0$ or $\mu_g > 0$, and that $\mu_{h^*} > 0$. In* RandProx-Minibatch, *suppose that $0 < \gamma < \frac{2}{L_f}$. Define the Lyapunov function, for every $t \geq 0$,*

$$\Psi^t := \frac{1}{\gamma} \left\| x^t - x^\star \right\|^2 + \frac{n}{k} \left( \gamma n + 2\mu_{h^*} \right) \sum_{i=1}^{n} \left\| u_i^t - u_i^\star \right\|^2, \tag{25}$$

*where $x^\star$ and $(u_i^\star)_{i=1}^{n}$ are the unique solutions to (23) and (24), respectively. Then* RandProx-Minibatch *converges linearly: for every $t \geq 0$, $\mathbb{E}[\Psi^t] \leq c^t \Psi^0$, where*

$$c := \max \left( \frac{(1-\gamma\mu_f)^2}{1+\gamma\mu_g}, \frac{(\gamma L_f - 1)^2}{1+\gamma\mu_g}, 1 - \frac{2k\mu_{h^*}}{n(\gamma n + 2\mu_{h^*})} \right). \tag{26}$$

*Also, $(x^t)_{t \in \mathbb{N}}$ and $(\hat{x}^t)_{t \in \mathbb{N}}$ both converge to $x^\star$ and $(u_i^t)_{t \in \mathbb{N}}$ converges to $u_i^\star$, $\forall i$, almost surely.*

RandProx-Minibatch with $k = 1$ becomes the Stochastic Decoupling Method (SDM) proposed in Mishchenko & Richtárik (2019), where strong convexity of $g$ is not exploited, but similar guarantees are derived as in Theorem 5 if $\mu_g = 0$. Linear convergence of SDM is also proved in Mishchenko & Richtárik (2019) in conditions related to ours in Theorems 2 and 4. Thus, RandProx-Minibatch extends SDM to larger minibatch size $k$ and exploits possible strong convexity of $g$.

When $f = 0$ and $g = 0$, SDM further simplifies to Point-SAGA (Defazio, 2016). In that case, our results do not apply directly, since there is no strong convexity in $f$ and $g$ any more, but when minimizing the average of functions $h_i$, with each function supposed to be $L$-smooth and $\mu$-strongly convex, for some $L \geq \mu > 0$, we can transfer the strong convexity to $g$ by subtracting $\frac{\mu}{2}\|\cdot\|^2$ to each $h_i$ and setting $g = \frac{\mu}{2}\|\cdot\|^2$. This does not change the problem and the algorithm but our Theorem 5 now applies, and with the right choice of $\gamma$, we recover the result in Defazio (2016), that the asymptotic complexity of Point-SAGA to reach $\epsilon$-accuracy is $\mathcal{O}\left( \left( n + \sqrt{\frac{nL}{\mu}} \right) \log \frac{1}{\epsilon} \right)$, which is conjectured to be optimal.

Thus, RandProx-Minibatch extends Point-SAGA to larger minibatch size and to the more general problem (23) with nonzero $f$ or $g$.

When $n = 1$, there is no randomness and SDM reverts to the DY algorithm discussed in Appendix G.

---

**Algorithm 7** SDM
(Mishchenko & Richtárik, 2019)

---

**input:** initial points $x^0 \in \mathcal{X}$, $(u_i^0)_{i=1}^n \in \mathcal{X}^n$;
stepsize $\gamma > 0$
$v^0 := \sum_{i=1}^n u_i^0$
**for** $t = 0, 1, \ldots$ **do**
 $\hat{x}^t := \mathrm{prox}_{\gamma g}\big(x^t - \gamma \nabla f(x^t) - \gamma v^t\big)$
 pick $i^t \in \{1, \ldots, n\}$ uniformly at random
 $x^{t+1} := \mathrm{prox}_{\gamma n h_i}(\gamma n u_{i^t}^t + \hat{x}^t)$
 $u_{i^t}^{t+1} := u_{i^t}^t + \frac{1}{\gamma n}(\hat{x}^t - x^{t+1})$
 for every $i \in \{1, \ldots, n\}\backslash\{i^t\}$, $u_i^{t+1} := u_i^t$
 $v^{t+1} := \sum_{i=1}^n u_i^{t+1}$ // $= v^t + u_{i^t}^{t+1} - u_{i^t}^t$
**end for**

---

**Algorithm 8** Point-SAGA
(Defazio, 2016)

---

**input:** initial points $x^0 \in \mathcal{X}$, $(u_i^0)_{i=1}^n \in \mathcal{X}^n$;
stepsize $\gamma > 0$
$v^0 := \sum_{i=1}^n u_i^0$
**for** $t = 0, 1, \ldots$ **do**
 $\hat{x}^t := x^t - \gamma v^t$
 pick $i^t \in \{1, \ldots, n\}$ uniformly at random
 $x^{t+1} := \mathrm{prox}_{\gamma n h_i}(\gamma n u_{i^t}^t + \hat{x}^t)$
 $u_{i^t}^{t+1} := u_{i^t}^t + \frac{1}{\gamma n}(\hat{x}^t - x^{t+1})$
 for every $i \in \{1, \ldots, n\}\backslash\{i^t\}$, $u_i^{t+1} := u_i^t$
 $v^{t+1} := \sum_{i=1}^n u_i^{t+1}$ // $= v^t + u_{i^t}^{t+1} - u_{i^t}^t$
**end for**

---

**Algorithm 9** RandProx-FL [new]

---

**input:** initial estimates $(x_i^0)_{i=1}^n \in \mathcal{X}^n$, $(u_i^0)_{i=1}^n \in \mathcal{X}^n$ such that $\sum_{i=1}^n u_i^0 = 0$; stepsize $\gamma > 0$; $\omega \geq 0$
**for** $t = 0, 1, \ldots$ **do**
 **for** $i = 1, \ldots, n$ at nodes in parallel **do**
  $\hat{x}_i^t := x_i^t - \gamma \nabla f_i(x_i^t) - \gamma u_i^t$
  $a_i^t := \mathcal{R}^t(\hat{x}_i^t)$
  // send compressed vector $a_i^t$ to master
 **end for**
 $a^t := \frac{1}{n}\sum_{i=1}^n a_i^t$ // aggregation at master
 // broadcast $a^t$ to all nodes
 **for** $i = 1, \ldots, n$ at nodes in parallel **do**
  $d_i^t := a_i^t - a^t$
  $u_i^{t+1} := u_i^t + \frac{1}{\gamma(1+\omega)^2} d_i^t$
  $x_i^{t+1} := \hat{x}_i^t - \frac{1}{1+\omega} d_i^t$
 **end for**
**end for**

---

## A.3 DISTRIBUTED AND FEDERATED LEARNING WITH COMPRESSION

We consider in this section distributed optimization within the client-server model, with a master node communicating back and forth with $n \geq 1$ parallel workers. This is particularly relevant for federated learning (FL) (Konečný et al., 2016; McMahan et al., 2017; Kairouz et al., 2021; Li et al., 2020), where a potentially huge number of devices, with their owners' data stored on each of them, are involved in the collaborative process of training a global machine learning model. The goal is to exploit the wealth of useful information lying in the heterogeneous data stored across the devices. Communication between the devices and the distant server, which can be costly and slow, is the main bottleneck in this framework. So, it is of primary importance to devise novel algorithmic strategies, which are efficient in terms of computation and communication complexities. A natural and widely used idea is to make use of (lossy) *compression*, to reduce the size of the communicated message (Alistarh et al., 2017; Wen et al., 2017; Wangni et al., 2018; Khaled & Richtárik, 2019; Albasyoni et al., 2020; Basu et al., 2020; Dutta et al., 2020; Sattler et al., 2020; Xu et al., 2021). Another popular idea is to make use of *local steps* (McMahan et al., 2017; Khaled et al., 2019; Stich, 2019; Khaled et al., 2020a; Malinovsky et al., 2020; Woodworth et al., 2020; Karimireddy et al., 2020; Gorbunov et al., 2021; Mishchenko et al., 2022); that is, communication with the server does not occur at every iteration but only every few iterations, for instance communication is triggered randomly with a small probability at every iteration. Between communication rounds, the workers perform multiple local steps independently, based on their local objectives. Our proposed algorithm RandProx-FL unifies the two strategies, in the sense that depending on the choice of the randomization

process $\mathcal{R}^t$, we obtain a method with local steps or with compression, or both. The combination of local training and compression has been further investigated in our follow-up work (Condat et al., 2022a), and partial participation in Condat et al. (2023b).

Thus, we consider the problem

$$\text{Find } x^\star \in \underset{x \in \mathbb{R}^d}{\arg\min} \left( \sum_{i=1}^{n} f_i(x) \right), \tag{27}$$

where $d \geq 1$ is the model dimension and $n \geq 1$ is the number of parallel workers, each having its own objective function $f_i$. Every function $f_i : \mathbb{R}^d \to \mathbb{R}$ is $\mu$-strongly convex and $L$-smooth, for some $L \geq \mu > 0$. We define $\kappa \coloneqq L/\mu$.

Now, we can observe that (27) can be recast as (1) with $K = \text{Id}, \mathcal{U} = \mathcal{X}, g = 0$; that is, as the minimization of $f + h$, as studied in Section 4.1, with

$$\mathcal{X} = (\mathbb{R}^d)^n, \quad f : x = (x_i)_{i=1}^n \mapsto \sum_{i=1}^{n} f_i(x_i), \tag{28}$$

$$h : x = (x_i)_{i=1}^n \mapsto (0 \text{ if } x_1 = \cdots = x_n, +\infty \text{ otherwise}). \tag{29}$$

We note that $f$ is $\mu$-strongly convex and $L$-smooth, and $\mu_{h^*} = 0$. Making these substitutions in RandProx-FB yields RandProx-FL, a distributed algorithm well suited for FL, shown above. In RandProx-FL, randomization takes the form of *linear* random unbiased operators $\mathcal{R}^t$ applied to the vectors sent to the server. Note that at every iteration, the same operator $\mathcal{R}^t$ is applied at every node; that is, its randomness is shared. We can easily check that RandProx-FL is an instance of RandProx-FB, because of the linearity of the $\mathcal{R}^t$ and because the property $\sum_{i=1}^{n} u_i^t = 0$ is maintained at every iteration. Formally, $\mathcal{R}^t$ applied as a whole in RandProx-FB consists of $n$ copies of $\mathcal{R}^t$ applied individually at every node in RandProx-FL, that is why we keep the same notation; in particular, the value of $\omega$ is the same in both interpretations.

Interestingly, in RandProx-FL, information about the functions $f_i$ or their gradients is never communicated and is exploited completely locally. This is ideal in terms of privacy.

As an application of Theorem 3, we obtain:

**Theorem 10.** *In RandProx-FL, suppose that $0 < \gamma < \frac{2}{L_f}$. Define the Lyapunov function, for every $t \geq 0$,*

$$\Psi^t \coloneqq \sum_{i=1}^{n} \left( \frac{1}{\gamma} \left\| x_i^t - x^\star \right\|^2 + \gamma(1 + \omega)^2 \left\| u_i^t - u_i^\star \right\|^2 \right), \tag{30}$$

*where $x^\star$ is the unique solution of (27) and $u_i^\star \coloneqq -\nabla f_i(x^\star)$. Then RandProx-FL converges linearly: for every $t \geq 0$, $\mathbb{E}[\Psi^t] \leq c^t \Psi^0$, where*

$$c \coloneqq \max \left( (1 - \gamma\mu_f)^2, (\gamma L_f - 1)^2, 1 - \frac{1}{(1 + \omega)^2} \right) < 1. \tag{31}$$

*Also, the $(x_i^t)_{t \in \mathbb{N}}$ and $(\hat{x}_i^t)_{t \in \mathbb{N}}$ all converge to $x^\star$ and every $(u_i^t)_{t \in \mathbb{N}}$ converges to $u_i^\star$, almost surely.*

If $\mathcal{R}^t$ is the Bernoulli compressor we have seen before in (6) and in Section A.1, RandProx-FL reverts to the Scaffnew algorithm proposed in Mishchenko et al. (2022), which communicates at every iteration with probability $p \in (0, 1]$ and performs in average $1/p$ local steps between successive communication rounds. We have $\omega = \frac{1}{p} - 1$. The analysis of Scaffnew in Theorem 10 is the same as in Mishchenko et al. (2022). With $\gamma = \frac{1}{L}$, the iteration complexity of Scaffnew is $\mathcal{O}\big( (\kappa + \frac{1}{p^2}) \log \frac{1}{\epsilon} \big)$, and since the algorithm communicates with probability $p$, its average communication complexity is $\mathcal{O}\big( (p\kappa + \frac{1}{p}) \log \frac{1}{\epsilon} \big)$. In particular, with $p = \frac{1}{\sqrt{\kappa}}$, the average communication complexity of Scaffnew is $\mathcal{O}\big( \sqrt{\kappa} \log \frac{1}{\epsilon} \big)$.

We now propose a new algorithm with compressed communication: in RandProx-FL we choose, for every $t \geq 0$, $\mathcal{R}^t$ as the well-known rand-$k$ compressor, for some $k \in \{1, \ldots, d\}$: $\mathcal{R}^t$ multiplies $k$ coordinates, chosen uniformly at random, of its vector argument by $d/k$ and sets the other ones to zero. We have $\omega = \frac{d}{k} - 1$. The iteration complexity with $\gamma = \frac{1}{L}$ is $\mathcal{O}\big( (\kappa + \frac{d^2}{k^2}) \log \frac{1}{\epsilon} \big)$ and the

communication complexity, in terms of average number of floats sent by every worker to the master, is $\mathcal{O}\big((k\kappa + \frac{d^2}{k})\log\frac{1}{\epsilon}\big)$, since $k$ floats are sent by every worker at every iteration. Thus, by choosing $k = \lceil d/\sqrt{\kappa}\rceil$, as long as $d \geq \sqrt{\kappa}$, the communication complexity in terms of floats is $\mathcal{O}\big(d\sqrt{\kappa}\log\frac{1}{\epsilon}\big)$; this is the same as the one of Scaffnew with $\gamma = \frac{1}{L}$ and $p = \frac{1}{\sqrt{\kappa}}$, but RandProx-FL with rand-$k$ compressors removes the necessity to communicate full $d$-dimensional vectors periodically.

## B  CONTRACTION OF GRADIENT DESCENT

**Lemma 1.** *For every $\gamma > 0$, the gradient descent operator $\mathrm{Id} - \gamma\nabla f$ is $c_\gamma$-Lipschitz continuous, with $c_\gamma := \max(1 - \gamma\mu_f, \gamma L_f - 1)$. That is, for every $(x, x') \in \mathcal{X}^2$,*

$$\|(\mathrm{Id} - \gamma\nabla f)x - (\mathrm{Id} - \gamma\nabla f)x'\| \leq c_\gamma \|x - x'\|.$$

*Proof* Let $(x, x') \in \mathcal{X}^2$. By cocoercivity of $\nabla f - \mu_f \mathrm{Id}$, we have (Bubeck, 2015, Lemma 3.11) $\langle\nabla f(x) - \nabla f(x'), x - x'\rangle \geq \frac{L_f \mu_f}{L_f + \mu_f}\|x - x'\|^2 + \frac{1}{L_f + \mu_f}\|\nabla f(x) - \nabla f(x')\|^2$. Hence,

$$\|(\mathrm{Id} - \gamma\nabla f)x - (\mathrm{Id} - \gamma\nabla f)x'\|^2 \leq \big(1 - \tfrac{2\gamma L_f \mu_f}{L_f + \mu_f}\big)\|x - x'\|^2$$
$$+ \big(\gamma^2 - \tfrac{2\gamma}{L_f + \mu_f}\big)\|\nabla f(x) - \nabla f(x')\|^2.$$

Thus, if $\gamma \leq \frac{2}{L_f + \mu_f}$, since $\|\nabla f(x) - \nabla f(x')\| \geq \mu_f\|x - x'\|$,

$$\|(\mathrm{Id} - \gamma\nabla f)x - (\mathrm{Id} - \gamma\nabla f)x'\|^2 \leq \Big(1 - \tfrac{2\gamma L_f \mu_f}{L_f + \mu_f} + (\gamma^2 - \tfrac{2\gamma}{L_f + \mu_f})\mu_f^2\Big)\|x - x'\|^2$$
$$= (1 - \gamma\mu_f)^2\|x - x'\|^2.$$

On the other hand, if $\gamma \geq \frac{2}{L_f + \mu_f}$, since $\|\nabla f(x) - \nabla f(x')\| \leq L_f\|x - x'\|$,

$$\|(\mathrm{Id} - \gamma\nabla f)x - (\mathrm{Id} - \gamma\nabla f)x'\|^2 \leq \Big(1 - \tfrac{2\gamma L_f \mu_f}{L_f + \mu_f} + (\gamma^2 - \tfrac{2\gamma}{L_f + \mu_f})L_f^2\Big)\|x - x'\|^2$$
$$= (\gamma L_f - 1)^2\|x - x'\|^2.$$

Since $\max(1 - \gamma\mu_f, \gamma L_f - 1) = (1 - \gamma\mu_f$ if $\gamma \leq \frac{2}{L_f + \mu_f}, \gamma L_f - 1$ otherwise$) \geq 0$, we arrive at the given expression of $c_\gamma$. $\qquad\square$

We note that if $\gamma < \frac{2}{L_f}$ and $\mu_f > 0$, $c_\gamma < 1$.

## C  PROOF OF THEOREM 1

Let $t \in \mathbb{N}$. Let $p^t \in \partial g(\hat{x}^t)$ be such that $\hat{x}^t = x^t - \gamma\nabla f(x^t) - \gamma p^t - \gamma K^* u^t$; $p^t$ exists and is unique, by properties of the proximity operator. We also define $p^\star := -\nabla f(x^\star) - K^* u^\star$; we have $p^\star \in \partial g(x^\star)$. Let $q^t := p^t - \mu_g\hat{x}^t$ and $q^\star := p^\star - \mu_g x^\star$. We have $(1 + \gamma\mu_g)\hat{x}^t = x^t - \gamma\nabla f(x^t) - \gamma q^t - \gamma K^* u^t$. Let $w^t := x^t - \gamma\nabla f(x^t)$ and $w^\star := x^\star - \gamma\nabla f(x^\star)$.

Using $\hat{u}^{t+1}$ defined in (9), we have

$$\mathbb{E}\Big[\big\|x^{t+1} - x^\star\big\|^2 \mid \mathcal{F}_t\Big] = \big\|\mathbb{E}[x^{t+1} \mid \mathcal{F}_t] - x^\star\big\|^2 + \mathbb{E}\Big[\big\|x^{t+1} - \mathbb{E}[x^{t+1} \mid \mathcal{F}_t]\big\|^2 \mid \mathcal{F}_t\Big]$$
$$\leq \big\|\hat{x}^t - x^\star - \gamma K^*(\hat{u}^{t+1} - u^t)\big\|^2 + \gamma^2\omega_{\mathrm{ran}}\big\|\hat{u}^{t+1} - u^t\big\|^2$$
$$- \gamma^2\zeta\big\|K^*(\hat{u}^{t+1} - u^t)\big\|^2.$$

Moreover,

$$
\begin{aligned}
\left\|\hat{x}^t - x^\star - \gamma K^*(\hat{u}^{t+1} - u^t)\right\|^2 &= \left\|\hat{x}^t - x^\star\right\|^2 + \gamma^2 \left\|K^*(\hat{u}^{t+1} - u^t)\right\|^2 \\
&\quad - 2\gamma\langle \hat{x}^t - x^\star, K^*(\hat{u}^{t+1} - u^t)\rangle \\
&\leq (1 + \gamma\mu_g)\left\|\hat{x}^t - x^\star\right\|^2 + \gamma^2 \left\|K^*(\hat{u}^{t+1} - u^t)\right\|^2 \\
&\quad - 2\gamma\langle \hat{x}^t - x^\star, K^*(\hat{u}^{t+1} - u^\star)\rangle + 2\gamma\langle \hat{x}^t - x^\star, K^*(u^t - u^\star)\rangle \\
&= \langle w^t - w^\star - \gamma(q^t - q^\star) - \gamma K^*(u^t - u^\star), \hat{x}^t - x^\star\rangle \\
&\quad + \gamma^2 \left\|K^*(\hat{u}^{t+1} - u^t)\right\|^2 \\
&\quad - 2\gamma\langle \hat{x}^t - x^\star, K^*(\hat{u}^{t+1} - u^\star)\rangle + 2\gamma\langle \hat{x}^t - x^\star, K^*(u^t - u^\star)\rangle \\
&= -2\gamma\langle q^t - q^\star, \hat{x}^t - x^\star\rangle \\
&\quad + \langle w^t - w^\star + \gamma(q^t - q^\star) + \gamma K^*(u^t - u^\star), \hat{x}^t - x^\star\rangle \\
&\quad + \gamma^2 \left\|K^*(\hat{u}^{t+1} - u^t)\right\|^2 - 2\gamma\langle \hat{x}^t - x^\star, K^*(\hat{u}^{t+1} - u^\star)\rangle \\
&= -2\gamma\langle q^t - q^\star, \hat{x}^t - x^\star\rangle \\
&\quad + \frac{1}{1 + \gamma\mu_g}\langle w^t - w^\star + \gamma(q^t - q^\star) + \gamma K^*(u^t - u^\star), \\
&\qquad w^t - w^\star - \gamma(q^t - q^\star) - \gamma K^*(u^t - u^\star)\rangle \\
&\quad + \gamma^2 \left\|K^*(\hat{u}^{t+1} - u^t)\right\|^2 - 2\gamma\langle \hat{x}^t - x^\star, K^*(\hat{u}^{t+1} - u^\star)\rangle \\
&= -2\gamma\langle q^t - q^\star, \hat{x}^t - x^\star\rangle + \frac{1}{1 + \gamma\mu_g}\left\|w^t - w^\star\right\|^2 \\
&\quad - \frac{\gamma^2}{1 + \gamma\mu_g}\left\|q^t - q^\star + K^*(u^t - u^\star)\right\|^2 \\
&\quad + \gamma^2 \left\|K^*(\hat{u}^{t+1} - u^t)\right\|^2 - 2\gamma\langle \hat{x}^t - x^\star, K^*(\hat{u}^{t+1} - u^\star)\rangle.
\end{aligned}
$$

We have $\langle q^t - q^\star, \hat{x}^t - x^\star\rangle \geq 0$. Hence,

$$
\begin{aligned}
\left\|\hat{x}^t - x^\star - \gamma K^*(\hat{u}^{t+1} - u^t)\right\|^2 &\leq \frac{1}{1 + \gamma\mu_g}\left\|w^t - w^\star\right\|^2 - \frac{\gamma^2}{1 + \gamma\mu_g}\left\|q^t - q^\star + K^*(u^t - u^\star)\right\|^2 \\
&\quad + \gamma^2 \left\|K^*(\hat{u}^{t+1} - u^t)\right\|^2 - 2\gamma\langle \hat{x}^t - x^\star, K^*(\hat{u}^{t+1} - u^\star)\rangle,
\end{aligned}
$$

so that

$$
\begin{aligned}
\mathbb{E}\left[\left\|x^{t+1} - x^\star\right\|^2 \mid \mathcal{F}_t\right] &\leq \frac{1}{1 + \gamma\mu_g}\left\|w^t - w^\star\right\|^2 - \frac{\gamma^2}{1 + \gamma\mu_g}\left\|q^t - q^\star + K^*(u^t - u^\star)\right\|^2 \\
&\quad + \gamma^2(1 - \zeta)\left\|K^*(\hat{u}^{t+1} - u^t)\right\|^2 - 2\gamma\langle \hat{x}^t - x^\star, K^*(\hat{u}^{t+1} - u^\star)\rangle \\
&\quad + \gamma^2 \omega_{\mathrm{ran}}\left\|\hat{u}^{t+1} - u^t\right\|^2.
\end{aligned}
$$

On the other hand,

$$
\begin{aligned}
\mathbb{E}\left[\left\|u^{t+1} - u^\star\right\|^2 \mid \mathcal{F}_t\right] &\leq \left\|u^t - u^\star + \frac{1}{1 + \omega}\left(\hat{u}^{t+1} - u^t\right)\right\|^2 + \frac{\omega}{(1 + \omega)^2}\left\|\hat{u}^{t+1} - u^t\right\|^2 \\
&= \frac{\omega^2}{(1 + \omega)^2}\left\|u^t - u^\star\right\|^2 + \frac{1}{(1 + \omega)^2}\left\|\hat{u}^{t+1} - u^\star\right\|^2 \\
&\quad + \frac{2\omega}{(1 + \omega)^2}\langle u^t - u^\star, \hat{u}^{t+1} - u^\star\rangle + \frac{\omega}{(1 + \omega)^2}\left\|\hat{u}^{t+1} - u^\star\right\|^2 \\
&\quad + \frac{\omega}{(1 + \omega)^2}\left\|u^t - u^\star\right\|^2 - \frac{2\omega}{(1 + \omega)^2}\langle u^t - u^\star, \hat{u}^{t+1} - u^\star\rangle \\
&= \frac{1}{1 + \omega}\left\|\hat{u}^{t+1} - u^\star\right\|^2 + \frac{\omega}{1 + \omega}\left\|u^t - u^\star\right\|^2. \quad\quad (32)
\end{aligned}
$$

Let $s^{t+1} \in \partial h^*(\hat{u}^{t+1})$ be such that $\hat{u}^{t+1} = u^t + \tau K \hat{x}^t - \tau s^{t+1}$; $s^{t+1}$ exists and is unique. We also define $s^\star := K x^\star$; we have $s^\star \in \partial h^*(u^\star)$. Therefore,

$$
\begin{aligned}
\left\| \hat{u}^{t+1} - u^\star \right\|^2 &= \left\| (u^t - u^\star) + (\hat{u}^{t+1} - u^t) \right\|^2 \\
&= \left\| u^t - u^\star \right\|^2 + \left\| \hat{u}^{t+1} - u^t \right\|^2 + 2\langle u^t - u^\star, \hat{u}^{t+1} - u^t \rangle \\
&= \left\| u^t - u^\star \right\|^2 + 2\langle \hat{u}^{t+1} - u^\star, \hat{u}^{t+1} - u^t \rangle - \left\| \hat{u}^{t+1} - u^t \right\|^2 \\
&= \left\| u^t - u^\star \right\|^2 - \left\| \hat{u}^{t+1} - u^t \right\|^2 + 2\tau \langle \hat{u}^{t+1} - u^\star, K(\hat{x}^t - x^\star) \rangle \\
&\quad - 2\tau \langle \hat{u}^{t+1} - u^\star, s^{t+1} - s^\star \rangle.
\end{aligned}
$$

Hence,

$$
\begin{aligned}
\frac{1}{\gamma} \mathbb{E}\Big[ \left\| x^{t+1} - x^\star \right\|^2 & \mid \mathcal{F}_t \Big] + \frac{1+\omega}{\tau} \mathbb{E}\Big[ \left\| u^{t+1} - u^\star \right\|^2 \mid \mathcal{F}_t \Big] \\
&\leq \frac{1}{\gamma(1+\gamma\mu_g)} \left\| w^t - w^\star \right\|^2 - \frac{\gamma}{1+\gamma\mu_g} \left\| q^t - q^\star + K^*(u^t - u^\star) \right\|^2 \\
&\quad + \gamma(1-\zeta) \left\| K^*(\hat{u}^{t+1} - u^t) \right\|^2 - 2\langle \hat{x}^t - x^\star, K^*(\hat{u}^{t+1} - u^\star) \rangle \\
&\quad + \gamma\omega_{\mathrm{ran}} \left\| \hat{u}^{t+1} - u^t \right\|^2 + \frac{1}{\tau} \left\| u^t - u^\star \right\|^2 - \frac{1}{\tau} \left\| \hat{u}^{t+1} - u^t \right\|^2 \\
&\quad + 2\langle \hat{u}^{t+1} - u^\star, K(\hat{x}^t - x^\star) \rangle - 2\langle \hat{u}^{t+1} - u^\star, s^{t+1} - s^\star \rangle \\
&\quad + \frac{\omega}{\tau} \left\| u^t - u^\star \right\|^2 \\
&\leq \frac{1}{\gamma(1+\gamma\mu_g)} \left\| w^t - w^\star \right\|^2 - \frac{\gamma}{1+\gamma\mu_g} \left\| q^t - q^\star + K^*(u^t - u^\star) \right\|^2 \\
&\quad + \frac{1+\omega}{\tau} \left\| u^t - u^\star \right\|^2 + \left( \gamma((1-\zeta)\|K\|^2 + \omega_{\mathrm{ran}}) - \frac{1}{\tau} \right) \left\| \hat{u}^{t+1} - u^t \right\|^2 \\
&\quad - 2\langle \hat{u}^{t+1} - u^\star, s^{t+1} - s^\star \rangle \\
&\leq \frac{1}{\gamma(1+\gamma\mu_g)} \left\| w^t - w^\star \right\|^2 - \frac{\gamma}{1+\gamma\mu_g} \left\| q^t - q^\star + K^*(u^t - u^\star) \right\|^2 \\
&\quad + \frac{1+\omega}{\tau} \left\| u^t - u^\star \right\|^2 - 2\langle \hat{u}^{t+1} - u^\star, s^{t+1} - s^\star \rangle.
\end{aligned}
$$

By $\mu_{h^*}$-strong monotonicity of $\partial h^*$, $\langle \hat{u}^{t+1} - u^\star, s^{t+1} - s^\star \rangle \geq \mu_{h^*} \left\| \hat{u}^{t+1} - u^\star \right\|^2$, and using (32),

$$
\langle \hat{u}^{t+1} - u^\star, s^{t+1} - s^\star \rangle \geq \mu_{h^*} \left( (1+\omega)\mathbb{E}\Big[ \left\| u^{t+1} - u^\star \right\|^2 \mid \mathcal{F}_t \Big] - \omega \left\| u^t - u^\star \right\|^2 \right).
$$

Hence,

$$
\begin{aligned}
\frac{1}{\gamma} \mathbb{E}\Big[ \left\| x^{t+1} - x^\star \right\|^2 \mid \mathcal{F}_t \Big] &+ (1+\omega)\left( \frac{1}{\tau} + 2\mu_{h^*} \right) \mathbb{E}\Big[ \left\| u^{t+1} - u^\star \right\|^2 \mid \mathcal{F}_t \Big] \\
&\leq \frac{1}{\gamma(1+\gamma\mu_g)} \left\| w^t - w^\star \right\|^2 - \frac{\gamma}{1+\gamma\mu_g} \left\| q^t - q^\star + K^*(u^t - u^\star) \right\|^2 \\
&\quad + \left( \frac{1+\omega}{\tau} + 2\omega\mu_{h^*} \right) \left\| u^t - u^\star \right\|^2.
\end{aligned} \tag{33}
$$

After Lemma 1,

$$
\begin{aligned}
\left\| w^t - w^\star \right\|^2 &= \left\| (\mathrm{Id} - \gamma\nabla f)x^t - (\mathrm{Id} - \gamma\nabla f)x^\star \right\|^2 \\
&\leq \max(1 - \gamma\mu_f, \gamma L_f - 1)^2 \left\| x^t - x^\star \right\|^2.
\end{aligned}
$$

Plugging this inequality in (33) yields

$$
\begin{aligned}
\mathbb{E}\big[ \Psi^{t+1} \mid \mathcal{F}_t \big] &\leq \frac{1}{\gamma(1+\gamma\mu_g)} \max(1 - \gamma\mu_f, \gamma L_f - 1)^2 \left\| x^t - x^\star \right\|^2 \\
&\quad + \left( \frac{1+\omega}{\tau} + 2\omega\mu_{h^*} \right) \left\| u^t - u^\star \right\|^2 - \frac{\gamma}{1+\gamma\mu_g} \left\| q^t - q^\star + K^*(u^t - u^\star) \right\|^2.
\end{aligned} \tag{34}
$$

Ignoring the last term in (34), we obtain:

$$\mathbb{E}\big[\Psi^{t+1} \mid \mathcal{F}_t\big] \leq \max\left(\frac{(1-\gamma\mu_f)^2}{1+\gamma\mu_g}, \frac{(\gamma L_f - 1)^2}{1+\gamma\mu_g}, 1 - \frac{2\tau\mu_{h^*}}{(1+\omega)(1+2\tau\mu_{h^*})}\right)\Psi^t. \quad (35)$$

Using the tower rule, we can unroll the recursion in (35) to obtain the unconditional expectation of $\Psi^{t+1}$. Since $\mathbb{E}[\Psi^t] \to 0$, we have $\mathbb{E}\big[\|x^t - x^\star\|^2\big] \to 0$ and $\mathbb{E}\big[\|u^t - u^\star\|^2\big] \to 0$. Moreover, using classical results on supermartingale convergence (Bertsekas, 2015, Proposition A.4.5), it follows from (35) that $\Psi^t \to 0$ almost surely. Almost sure convergence of $x^t$ and $u^t$ follows. Finally, by Lipschitz continuity of $\nabla f$, $K^*$, $\mathrm{prox}_g$, we can upper bound $\|\hat{x}^t - x^\star\|^2$ by a linear combination of $\|x^t - x^\star\|^2$ and $\|u^t - u^\star\|^2$. It follows that $\mathbb{E}\big[\|\hat{x}^t - x^\star\|^2\big] \to 0$ linearly with the same rate $c$ and that $\hat{x}^t \to x^\star$ almost surely, as well. $\qquad\square$

## D  PROOF OF THEOREM 2

Let us go back to (34). Since $g = 0$, we have $q^t = q^\star = 0$ and $\mu_g = 0$, so that

$$\mathbb{E}\big[\Psi^{t+1} \mid \mathcal{F}_t\big] \leq \frac{1}{\gamma}\max(1-\gamma\mu_f, \gamma L_f - 1)^2 \big\|x^t - x^\star\big\|^2 + \left(\frac{1+\omega}{\tau} + 2\omega\mu_{h^*}\right)\big\|u^t - u^\star\big\|^2$$
$$- \gamma\big\|K^*(u^t - u^\star)\big\|^2.$$

We have $\|K^*(u^t - u^\star)\|^2 \geq \lambda_{\min}(KK^*)\|u^t - u^\star\|^2$. This yields

$$\mathbb{E}\big[\Psi^{t+1} \mid \mathcal{F}_t\big] \leq \frac{1}{\gamma}\max(1-\gamma\mu_f, \gamma L_f - 1)^2 \big\|x^t - x^\star\big\|^2$$
$$+ \left(\frac{1+\omega}{\tau} + 2\omega\mu_{h^*} - \gamma\lambda_{\min}(KK^*)\right)\big\|u^t - u^\star\big\|^2$$
$$\leq \max\left((1-\gamma\mu_f)^2, (\gamma L_f - 1)^2, 1 - \frac{2\tau\mu_{h^*} + \gamma\tau\lambda_{\min}(KK^*)}{(1+\omega)(1+2\tau\mu_{h^*})}\right)\Psi^t. \quad (36)$$

The end of the proof is the same as the one of Theorem 1. $\qquad\square$

Let us add here a remark on the PAPC algorithm, which is the particular case of RandProx when $\omega = 0$, in the conditions of Theorem 2:

**Remark 2** (PAPC vs. proximal gradient descent on the dual problem)  If $\mu_f > 0$, $f^*$ is $\mu^{-1}$-smooth and $L_f^{-1}$-strongly convex. Then $f^* \circ -K^*$ is $\mu_f^{-1}\|K\|^2$-smooth and $L_f^{-1}\lambda_{\min}(KK^*)$-strongly convex. So, if $\nabla f^*$ is computable, one can apply the proximal gradient algorithm on the dual problem (2), which iterates $u^{t+1} = \mathrm{prox}_{\tau h^*}\big(u^t + \tau K\nabla f^*(-K^*u^t)\big)$, with $\tau \in \big(0, \frac{2\mu_f}{\|K\|^2}\big)$. If $\lambda_{\min}(KK^*) > 0$, this algorithm converges linearly: $\|u^{t+1} - u^\star\|^2 \leq c^2\|u^t - u^\star\|^2$ with $c = \max\big(1 - \tau L_f^{-1}\lambda_{\min}(KK^*), \tau\mu_f^{-1}\|K\|^2 - 1\big)$. $c$ is smallest with $\tau = 2/\big(\mu_f^{-1}\|K\|^2 + L_f^{-1}\lambda_{\min}(KK^*)\big)$, in which case

$$c = \frac{1 - \frac{\mu_f}{L_f}\frac{\lambda_{\min}(KK^*)}{\|K\|^2}}{1 + \frac{\mu_f}{L_f}\frac{\lambda_{\min}(KK^*)}{\|K\|^2}}.$$

This is much worse than the rate of the PAPC algorithm, since it involves the product of the condition numbers $L_f/\mu_f$ and $\|K\|^2/\lambda_{\min}(KK^*)$, instead of their maximum. This is due to calling gradients of $f^* \circ -K^*$, whereas $f$ and $K$ are split, or decoupled, in the PAPC algorithm.

## E  PROOF OF THEOREM 4 AND FURTHER DISCUSSION

---

**Algorithm 10** RandPriLiCo [new]

   **input:** initial points $x^0 \in \mathcal{X}$, $v^0 \in \operatorname{ran}(W)$;
   stepsizes $\gamma > 0$, $\tau > 0$; $\omega \geq 0$
   **for** $t = 0, 1, \dots$ **do**
      $\hat{x}^t := x^t - \gamma \nabla f(x^t) - \gamma v^t$
      $d^{t+1} := \tau \mathcal{S}^t(W \hat{x}^t - a)$
      $v^{t+1} := v^t + \frac{1}{1+\omega} d^{t+1}$
      $x^{t+1} := \hat{x}^t - \gamma d^{t+1}$
   **end for**

---

We observe that in RandProx-LC and Theorem 4, it is as if the sequence $(u_0^t)_{t\in\mathbb{N}}$ had been computed by the following iteration, initialized with $x^0 \in \mathcal{X}$ and $u_0^0 := P_{\operatorname{ran}(K)}(u^0)$:

$$\left|\begin{array}{l} \hat{x}^t := x^t - \gamma \nabla f(x^t) - \gamma v^t \\ u_0^{t+1} := u_0^t + \frac{1}{1+\omega} P_{\operatorname{ran}(K)} \mathcal{R}^t\big(\tau(K\hat{x}^t - b)\big) \\ v^{t+1} := K^* u_0^{t+1} \\ x^{t+1} := \hat{x}^t - \gamma(1+\omega)(v^{t+1} - v^t) \end{array}\right. .$$

Then we remark that this is simply the iteration of RandProx, with $\mathcal{R}^t$ replaced by $\widetilde{\mathcal{R}}^t := P_{\operatorname{ran}(K)} \mathcal{R}^t$. Since its argument $r^t = \tau(K\hat{x}^t - b)$ is always in $\operatorname{ran}(K)$, $\widetilde{\mathcal{R}}^t$ is unbiased, and we have, for every $t \geq 0$,

$$\mathbb{E}\Big[\big\|\widetilde{\mathcal{R}}^t(r^t) - r^t\big\|^2 \mid \widetilde{\mathcal{F}}_t\Big] \leq \mathbb{E}\Big[\big\|\mathcal{R}^t(r^t) - r^t\big\|^2 \mid \widetilde{\mathcal{F}}_t\Big] \leq \omega \big\|r^t\big\|^2,$$

where $\widetilde{\mathcal{F}}_t$ the $\sigma$-algebra generated by the collection of random variables $(x^0, u_0^0), \dots, (x^t, u_0^t)$. Also, $\omega_{\operatorname{ran}}$ is unchanged. Therefore, the analysis of RandProx in Theorem 2 applies, with $u^t$ replaced by $u_0^t$ and $u^\star$ by $u_0^\star$. Now, for every $u \in \operatorname{ran}(K)$,

$$\|K^* u\|^2 \geq \lambda_{\min}^+(KK^*) \|u\|^2,$$

and using this lower bound in the proof of Theorem 2, with $\mu_{h^*} = 0$, we obtain Theorem 4. $\qquad\square$

Furthermore, the constraint $Kx = b$ is equivalent to the constraint $K^*Kx = K^*b$; so, let us consider problems where we are given $K^*K$ and not $K$ in the first place:

Let $W$ be a linear operator on $\mathcal{X}$, which is self-adjoint, i.e. $W^* = W$, and positive, i.e. $\langle Wx, x\rangle \geq 0$ for every $x \in \mathcal{X}$. Let $a \in \operatorname{ran}(W)$. We consider the linearly constrained minimization problem

$$\text{Find } x^\star \in \arg\min_{x\in\mathcal{X}} f(x) \quad \text{s.t.} \quad Wx = a. \tag{37}$$

Now, we let $\mathcal{U} := \mathcal{X}$ and $K = K^* := \sqrt{W}$, where $\sqrt{W}$ is the unique positive self-adjoint linear operator on $\mathcal{X}$ such that $\sqrt{W}\sqrt{W} = W$. Also, $b$ is defined as the unique element in $\operatorname{ran}(W) = \operatorname{ran}(K)$ such that $\sqrt{W}b = a$. Then (37) is equivalent to (17) and the dual problem is (18). We consider the Randomized Primal Linearly Constrained minimization algorithm (RandPriLiCo), shown above. We suppose that the stochastic operators $\mathcal{S}^t$ in RandPriLiCo satisfy, for every $t \geq 0$,

$$\mathbb{E}\Big[\mathcal{S}^t(r^t) \mid \widetilde{\mathcal{F}}_t\Big] = r^t \quad \text{and} \quad \mathbb{E}\Big[\big\|\mathcal{S}^t(r^t) - r^t\big\|^2 \mid \widetilde{\mathcal{F}}_t\Big] \leq \omega \big\|r^t\big\|^2, \tag{38}$$

for some $\omega \geq 0$, where $r^t := \tau W\hat{x}^t - \tau a$.

In addition, we suppose that the $\mathcal{S}^t$ commute with $\sqrt{W}$: for every $t \geq 0$ and $x \in \mathcal{X}$,

$$\sqrt{W}\mathcal{S}^t(x) = \mathcal{S}^t(\sqrt{W}x).$$

This is satisfied with the Bernoulli operators or some linear sketching operators, for instance. Then RandPriLiCo is equivalent to RandProx-LC, with $\mathcal{S}^t$ playing the role of $\mathcal{R}^t$ and $\omega_{\operatorname{ran}} = \|W\|\omega$, $\zeta = 0$. Applying Theorem 4 with these equivalences, we obtain:

**Algorithm 11** CP algorithm
(Chambolle & Pock, 2011)

> **input:** initial points $x^0 \in \mathcal{X}$, $u^0 \in \mathcal{U}$;
> stepsizes $\gamma > 0$, $\tau > 0$
> $\hat{x}^0 := \operatorname{prox}_{\gamma g}\big(x^0 - \gamma K^* u^0\big)$
> **for** $t = 0, 1, \ldots$ **do**
>    $u^{t+1} := \operatorname{prox}_{\tau h^*}\big(u^t + \tau K \hat{x}^t\big)$
>    // $x^{t+1} := \hat{x}^t - \gamma K^*(u^{t+1} - u^t)$
>    $\hat{x}^{t+1} := \operatorname{prox}_{\gamma g}\big(\hat{x}^t - \gamma K^*(2u^{t+1} - u^t)\big)$
> **end for**

**Algorithm 12** RandProx-CP [new]

> **input:** initial points $x^0 \in \mathcal{X}$, $u^0 \in \mathcal{U}$;
> stepsizes $\gamma > 0$, $\tau > 0$; $\omega \geq 0$
> $\hat{x}^0 := \operatorname{prox}_{\gamma g}\big(x^0 - \gamma K^* u^0\big)$
> **for** $t = 0, 1, \ldots$ **do**
>    $d^t := \mathcal{R}^t\big(\operatorname{prox}_{\tau h^*}(u^t + \tau K \hat{x}^t) - u^t\big)$
>    $u^{t+1} := u^t + \frac{1}{1+\omega} d^t$
>    // $x^{t+1} := \hat{x}^t - \gamma K^* d^t$
>    $\hat{x}^{t+1} := \operatorname{prox}_{\gamma g}\big(\hat{x}^t - \gamma K^*(u^{t+1} + d^t)\big)$
> **end for**

**Theorem 6.** *In the setting of (37), suppose that $\mu_f > 0$. In* RandPriLiCo, *suppose that $0 < \gamma < \frac{2}{L_f}$, $\tau > 0$ and $\gamma\tau\|W\|(1+\omega) \leq 1$. Define the Lyapunov function, for every $t \geq 0$,*

$$\Psi^t := \frac{1}{\gamma}\left\|x^t - x^\star\right\|^2 + \frac{1+\omega}{\tau}\left\|u_0^t - u_0^\star\right\|^2, \tag{39}$$

*where $u_0^t$ is the unique element in $\operatorname{ran}(W)$ such that $v^t = \sqrt{W}u_0^t$, $x^\star$ is the unique solution of (37) and $u_0^\star$ is the unique element in $\operatorname{ran}(W)$ such that $-\nabla f(x^\star) = \sqrt{W}u_0^\star$. Then* RandPriLiCo *converges linearly: for every $t \geq 0$,*

$$\mathbb{E}\big[\Psi^t\big] \leq c^t \Psi^0, \tag{40}$$

*where*

$$c := \max\left((1 - \gamma\mu_f)^2, (\gamma L_f - 1)^2, 1 - \frac{\gamma\tau\lambda_{\min}^+(W)}{1+\omega}\right) < 1. \tag{41}$$

*Also, $(x^t)_{t \in \mathbb{N}}$ and $(\hat{x}^t)_{t \in \mathbb{N}}$ both converge to $x^\star$ almost surely.*

RandPriLiCo can be applied to decentralized optimization, like in Kovalev et al. (2020); Salim et al. (2022a) but with randomized communication; we leave the detailed study of this setting for future work.

## F    PARTICULAR CASE $f = 0$: RANDOMIZED CHAMBOLLE–POCK ALGORITHM

In this section, we suppose that $f = 0$. The primal problem (1) becomes:

$$\text{Find } x^\star \in \arg\min_{x \in \mathcal{X}}\Big(g(x) + h(Kx)\Big), \tag{42}$$

and the dual problem (2) becomes:

$$\text{Find } u^\star \in \arg\min_{u \in \mathcal{U}}\Big(g^*(-K^*u) + h^*(u)\Big). \tag{43}$$

The PDDY algorithm becomes the Chambolle-Pock (CP), a.k.a. PDHG, algorithm (Chambolle & Pock, 2011), shown above. RandProx can be rewritten as RandProx-CP, shown above, too. In both algorithms, the variable $x^t$ is not needed any more and can be removed.

Since $f = 0$, $L_f > 0$ can be set arbitrarily close to zero, so that Theorem 1 can be rewritten as:

**Theorem 7.** *Suppose that $\mu_g > 0$ and $\mu_{h^*} > 0$. In* RandProx-CP, *suppose that $\gamma > 0$, $\tau > 0$, $\gamma\tau\big((1-\zeta)\|K\|^2 + \omega_{\mathrm{ran}}\big) \leq 1$. Define the Lyapunov function, for every $t \geq 0$,*

$$\Psi^t := \frac{1}{\gamma}\left\|x^t - x^\star\right\|^2 + (1+\omega)\left(\frac{1}{\tau} + 2\mu_{h^*}\right)\left\|u^t - u^\star\right\|^2, \tag{44}$$

*where $x^\star$ and $u^\star$ are the unique solutions to (42) and (43), respectively. Then* RandProx-CP *converges linearly: for every $t \geq 0$,*

$$\mathbb{E}\big[\Psi^t\big] \leq c^t \Psi^0, \tag{45}$$

---

**Algorithm 13** ADMM

  **input:** initial points $x^0 \in \mathcal{X}$, $u^0 \in \mathcal{U}$;
  stepsize $\gamma > 0$
  **for** $t = 0, 1, \ldots$ **do**
    $\hat{x}^t := \mathrm{prox}_{\gamma g}(x^t - \gamma u^t)$
    $x^{t+1} := \mathrm{prox}_{\gamma h}(\hat{x}^t + \gamma u^t)$
    $u^{t+1} := u^t + \frac{1}{\gamma}(\hat{x}^t - x^{t+1})$
  **end for**

---

**Algorithm 14** RandProx-ADMM [new]

  **input:** initial points $x^0 \in \mathcal{X}$, $u^0 \in \mathcal{U}$;
  stepsize $\gamma > 0$; $\omega \geq 0$
  **for** $t = 0, 1, \ldots$ **do**
    $\hat{x}^t := \mathrm{prox}_{\gamma g}(x^t - \gamma u^t)$
    $d^t := \mathcal{R}^t\big(\hat{x}^t - \mathrm{prox}_{\gamma(1+\omega)h}(\hat{x}^t + \gamma(1+\omega)u^t)\big)$
    $x^{t+1} := \hat{x}^t - \frac{1}{1+\omega}d^t$
    $u^{t+1} := u^t + \frac{1}{\gamma(1+\omega)^2}d^t$
  **end for**

---

*where*

$$c := \max\left(\frac{1}{1+\gamma\mu_g}, 1 - \frac{2\tau\mu_{h^*}}{(1+\omega)(1+2\tau\mu_{h^*})}\right) \tag{46}$$

$$= 1 - \min\left(\frac{\gamma\mu_g}{1+\gamma\mu_g}, \frac{2\tau\mu_{h^*}}{(1+\omega)(1+2\tau\mu_{h^*})}\right) < 1. \tag{47}$$

*Also, $(x^t)_{t\in\mathbb{N}}$ and $(\hat{x}^t)_{t\in\mathbb{N}}$ both converge to $x^\star$ and $(u^t)_{t\in\mathbb{N}}$ converges to $u^\star$, almost surely.*

It would be interesting to study whether the mechanism in the stochastic PDHG algorithm proposed in Chambolle et al. (2018) can be viewed as a particular case of RandProx-CP; we leave the analysis of this connection for future work. In any case, the strong convexity constants $\mu_g$ and $\mu_{h^*}$ need to be known in the linearly converging version of the stochastic PDHG algorithm, which is not the case here; this is an important advantage of RandProx-CP.

Now, let us look at the particular case $K = \mathrm{Id}$ in (42) and (43). The primal problem becomes:

$$\text{Find } x^\star \in \operatorname*{arg\,min}_{x\in\mathcal{X}} \Big(g(x) + h(x)\Big), \tag{48}$$

and the dual problem becomes:

$$\text{Find } u^\star \in \operatorname*{arg\,min}_{u\in\mathcal{U}} \Big(g^*(-u) + h^*(u)\Big). \tag{49}$$

When $K = \mathrm{Id}$, the CP algorithm with $\tau = \frac{1}{\gamma}$ reverts to the Douglas–Rachford algorithm, which is equivalent to the Alternating Direction Method of Multipliers (ADMM) (Boyd et al., 2011; Condat et al., 2023a), shown above. Therefore, in that case, with $\omega_{\mathrm{ran}} = \omega$, $\zeta = 0$ and $\tau = \frac{1}{\gamma(1+\omega)}$, RandProx-CP can be rewritten as RandProx-ADMM, shown above. Theorem 7 becomes:

**Theorem 8.** *Suppose that $\mu_g > 0$ and $\mu_{h^*} > 0$. In RandProx-ADMM, suppose that $\gamma > 0$. For every $t \geq 0$, define the Lyapunov function*

$$\Psi^t := \frac{1}{\gamma}\left\|x^t - x^\star\right\|^2 + (1+\omega)\big(\gamma(1+\omega) + 2\mu_{h^*}\big)\left\|u^t - u^\star\right\|^2, \tag{50}$$

*where $x^\star$ and $u^\star$ are the unique solutions to (48) and (49), respectively. Then RandProx-ADMM converges linearly: for every $t \geq 0$,*

$$\mathbb{E}\big[\Psi^t\big] \leq c^t\Psi^0, \tag{51}$$

*where*

$$c := \max\left(\frac{1}{1+\gamma\mu_g}, 1 - \frac{2\tau\mu_{h^*}}{(1+\omega)(1+2\tau\mu_{h^*})}\right) \tag{52}$$

$$= 1 - \min\left(\frac{\gamma\mu_g}{1+\gamma\mu_g}, \frac{2\tau\mu_{h^*}}{(1+\omega)(1+2\tau\mu_{h^*})}\right) < 1. \tag{53}$$

*Also, $(x^t)_{t\in\mathbb{N}}$ and $(\hat{x}^t)_{t\in\mathbb{N}}$ both converge to $x^\star$ and $(u^t)_{t\in\mathbb{N}}$ converges to $u^\star$, almost surely.*

---

**Algorithm 15** DY algorithm
(Davis & Yin, 2017)

---

**input:** initial points $x^0 \in \mathcal{X}$, $u^0 \in \mathcal{X}$;
stepsize $\gamma > 0$
**for** $t = 0, 1, \ldots$ **do**
  $\hat{x}^t := \mathrm{prox}_{\gamma g}\big(x^t - \gamma \nabla f(x^t) - \gamma u^t\big)$
  $x^{t+1} := \mathrm{prox}_{\gamma h}(\hat{x}^t + \gamma u^t)$
  $u^{t+1} := u^t + \frac{1}{\gamma}(\hat{x}^t - x^{t+1})$
**end for**

---

**Algorithm 16** RandProx-DY [new]

---

**input:** initial points $x^0 \in \mathcal{X}$, $u^0 \in \mathcal{X}$;
stepsize $\gamma > 0$; $\omega \geq 0$
**for** $t = 0, 1, \ldots$ **do**
  $\hat{x}^t := \mathrm{prox}_{\gamma g}\big(x^t - \gamma \nabla f(x^t) - \gamma u^t\big)$
  $d^t := \mathcal{R}^t\big(\hat{x}^t - \mathrm{prox}_{\gamma(1+\omega)h}(\hat{x}^t + \gamma(1+\omega)u^t)\big)$

  $x^{t+1} := \hat{x}^t - \frac{1}{1+\omega}d^t$
  $u^{t+1} := u^t + \frac{1}{\gamma(1+\omega)^2}d^t$
**end for**

---

# G   PARTICULAR CASE $K = \mathrm{Id}$: RANDOMIZED DAVIS–YIN ALGORITHM

After the particular case $g = 0$ discussed in Section 4.1 and the particular case $f = 0$ discussed in Section F, we discuss in this section the third particular case $K = \mathrm{Id}$ in (1) and (2). The primal problem becomes:

$$\text{Find } x^\star \in \arg\min_{x \in \mathcal{X}} \Big( f(x) + g(x) + h(x) \Big), \tag{54}$$

and the dual problem becomes:

$$\text{Find } u^\star \in \arg\min_{u \in \mathcal{U}} \Big( (f+g)^*(-u) + h^*(u) \Big). \tag{55}$$

When $K = \mathrm{Id}$, the PDDY algorithm with $\tau = \frac{1}{\gamma}$ reverts to the Davis–Yin (DY) algorithm (Davis & Yin, 2017), shown above. Therefore, in that case, with $\omega_{\mathrm{ran}} = \omega$, $\zeta = 0$ and $\tau = \frac{1}{\gamma(1+\omega)}$, RandProx can be rewritten as RandProx-DY, shown above, too. When $g = 0$, RandProx-DY reverts to RandProx-FB and when $f = 0$, RandProx-DY reverts to RandProx-ADMM; in other words, RandProx-DY generalizes RandProx-FB and RandProx-ADMM into a single algorithm. Theorem 1 yields:

**Theorem 9.** *Suppose that $\mu_f > 0$ or $\mu_g > 0$, and that $\mu_{h^*} > 0$. In RandProx-DY, suppose that $0 < \gamma < \frac{2}{L_f}$. For every $t \geq 0$, define the Lyapunov function,*

$$\Psi^t := \frac{1}{\gamma}\left\| x^t - x^\star \right\|^2 + (1+\omega)\big(\gamma(1+\omega) + 2\mu_{h^*}\big) \left\| u^t - u^\star \right\|^2, \tag{56}$$

*where $x^\star$ and $u^\star$ are the unique solutions to (54) and (55), respectively. Then RandProx-DY converges linearly: for every $t \geq 0$,*

$$\mathbb{E}\big[\Psi^t\big] \leq c^t \Psi^0, \tag{57}$$

*where*

$$c := \max\left( \frac{(1 - \gamma\mu_f)^2}{1 + \gamma\mu_g}, \frac{(\gamma L_f - 1)^2}{1 + \gamma\mu_g}, 1 - \frac{\frac{2}{\gamma}\mu_{h^*}}{(1+\omega)\big(1 + \omega + \frac{2}{\gamma}\mu_{h^*}\big)} \right) < 1. \tag{58}$$

*Also, $(x^t)_{t \in \mathbb{N}}$ and $(\hat{x}^t)_{t \in \mathbb{N}}$ both converge to $x^\star$ and $(u^t)_{t \in \mathbb{N}}$ converges to $u^\star$, almost surely.*

We note that in Theorem 9, $\mu_{h^*} > 0$ is required. It is only in the case $g = 0$, when RandProx-DY reverts to RandProx-FB, that one can apply Theorem 3, which does not require strong convexity of $h^*$.

# H   PROOF OF THEOREM 11

*Proof of Theorem 11* We have, for every $(x, x') \in \mathcal{X}^2$,

$$\|(\mathrm{Id} - \gamma\nabla f)x - (\mathrm{Id} - \gamma\nabla f)x'\|^2 = \|x - x'\|^2 - 2\gamma\langle \nabla f(x) - \nabla f(x'), x - x' \rangle$$
$$+ \gamma^2 \|\nabla f(x) - \nabla f(x')\|^2$$
$$\leq \|x - x'\|^2 - (2\gamma - \gamma^2 L_f)\langle \nabla f(x) - \nabla f(x'), x - x' \rangle,$$

where the second inequality follows from cocoercivity of the gradient. Moreover, for every $(x, x') \in \mathcal{X}^2$, $D_f(x, x') \leq \langle \nabla f(x) - \nabla f(x'), x - x' \rangle$. Therefore, in the proof of Theorem 1, for every primal–dual solution $(x^\star, u^\star)$ and $t \geq 0$, since $\|w^t - w^\star\|^2 = \|(\text{Id} - \gamma \nabla f)x^t - (\text{Id} - \gamma \nabla f)x^\star\|^2$, (33) yields

$$
\begin{aligned}
\mathbb{E}\big[\Psi^{t+1} \mid \mathcal{F}_t\big] \leq &\frac{1}{\gamma} \left\|x^t - x^\star\right\|^2 - (2 - \gamma L_f) D_f(x^t, x^\star) \\
&+ \left(\frac{1 + \omega}{\tau} + 2\omega\mu_{h^*}\right) \left\|u^t - u^\star\right\|^2 - \gamma \left\|q^t - q^\star + K^*(u^t - u^\star)\right\|^2.
\end{aligned}
$$

Ignoring the last term, this yields

$$
\begin{aligned}
\mathbb{E}\big[\Psi^{t+1} \mid \mathcal{F}_t\big] \leq &\frac{1}{\gamma} \left\|x^t - x^\star\right\|^2 + c(1 + \omega)\left(\frac{1}{\tau} + 2\mu_{h^*}\right) \left\|u^t - u^\star\right\|^2 \quad (59) \\
&- (2 - \gamma L_f) D_f(x^t, x^\star) \\
\leq &\Psi^t - (2 - \gamma L_f) D_f(x^t, x^\star), \quad (60)
\end{aligned}
$$

with $c = 1 - \frac{2\tau\mu_{h^*}}{(1+\omega)(1+2\tau\mu_{h^*})}$ in (59). Using classical results on supermartingale convergence (Bertsekas, 2015, Proposition A.4.5), it follows from (60) that $\Psi^t$ converges almost surely to a random variable $\Psi^\infty$ and that

$$
\sum_{t=0}^{\infty} D_f(x^t, x^\star) < +\infty \quad \text{almost surely.}
$$

Hence, $D_f(x^t, x^\star) \to 0$ almost surely. Moreover, for every $T \geq 0$,

$$
(2 - \gamma L_f) \sum_{t=0}^{T} \mathbb{E}\big[D_f(x^t, x^\star)\big] \leq \Psi^0 - \mathbb{E}\big[\Psi^{T+1}\big] \leq \Psi^0 \quad (61)
$$

and

$$
(2 - \gamma L_f) \sum_{t=0}^{\infty} \mathbb{E}\big[D_f(x^t, x^\star)\big] \leq \Psi^0.
$$

Therefore, $\mathbb{E}[D_f(x^t, x^\star)] \to 0$; that is, $D_f(x^t, x^\star) \to 0$ in quadratic mean.

The Bregman divergence is convex in its first argument, so that for every $T \geq 0$,

$$
D_f(\bar{x}^T, x^\star) \leq \frac{1}{T+1} \sum_{t=0}^{T} D_f(x^t, x^\star).
$$

Combining this last inequality with (61) yields

$$
(T + 1)(2 - \gamma L_f) \mathbb{E}\big[D_f(\bar{x}^T, x^\star)\big] \leq \Psi^0.
$$

Now, if $\mu_{h^*} > 0$, then $c < 1$ in (59), and since $\Psi^t$ converges almost surely to $\Psi^\infty$, it must be that $\mathbb{E}\left[\left\|u^t - u^\star\right\|^2\right] \to 0$. □

The counterpart of Theorem 2 in the convex case is:

**Theorem 12.** *Suppose that $g = 0$, and that $\lambda_{\min}(KK^*) > 0$ or $\mu_{h^*} > 0$. In* RandProx, *suppose that $0 < \gamma < \frac{2}{L_f}$, $\tau > 0$, and $\gamma\tau\big((1 - \zeta)\|K\|^2 + \omega_{\mathrm{ran}}\big) \leq 1$. Then there is a unique dual solution $u^\star$ to (2) and $(u^t)_{t \in \mathbb{N}}$ converges to $u^\star$, in quadratic mean.*

*Proof of Theorem 12* Considering the proof of Theorem 2, the same arguments as in the proof of Theorem 11 apply, with $c$ in (59) now equal to

$$
c = 1 - \frac{2\tau\mu_{h^*} + \gamma\tau\lambda_{\min}(KK^*)}{(1 + \omega)(1 + 2\tau\mu_{h^*})} < 1.
$$

Hence, $\mathbb{E}\left[\left\|u^t - u^\star\right\|^2\right] \to 0$. □

