# OpenReview forum: "RandProx: Primal-Dual Optimization Algorithms with Randomized Proximal Updates"
_ICLR.cc/2023/Conference — ICLR 2023 poster_

### Official Review · Reviewer_Lt4E · 2022-10-21

**Confidence:** 3
**Correctness:** 4
**Technical Novelty And Significance:** 1
**Empirical Novelty And Significance:** Not applicable
**Recommendation:** 3

**Clarity, Quality, Novelty And Reproducibility:**

- The novelty of this paper is limited. The algorithm proposed and studied by the paper is almost the same as the existing PDDY algorithm, except that the dual update is randomized.

- The advantage of randomization is not well supported. First, the paper doesn't give any practical scenario where the randomization is necessary or brings a significant improvement. Second, the convergence rate of the randomized algorithm is the same as the deterministic counterpart.

- It seems that the paper is compressed from a much longer version to suit the page limit without careful checking. For example, Theorem 11 follows right after Theorem 4 while Theorems 5-10 are missing. Yet Table 1 mentions Theorem 7-9.

- The paper looks like a stack of assumptions and theorems and doesn't provide many implications or insights from the Theorems. For example, Equation (7) states that the variance of $\mathcal{R}^t$ should be proportional to $\|r^t\|^2$. There is no discussion about what kind of randomization satisfies this assumption. For another example, Section 4.1 and 4.2 only states more Theorems about some special cases of Theorem 1 with few new insights. Similarly, Section 5 just introduces some basic concepts and lists two theorems.

- The paper doesn't have any numerical results supporting the applicability of the proposed algorithm.

**Strength And Weaknesses:**

**Strengh**

- The paper proposes an algorithm that introduces randomness in the dual update of the PDDY algorithm and proves the linear convergence under the strongly convex assumption.

**Weakness**

- It is not clear why the problem $f(x)+g(x)+h(Kx)$ is important to study. I suggest introducing more background and motivation for this problem.

- It is not clear why adding noise in the dual step is a good idea. It seems that the only benefit is that the cost per iteration might decrease if the randomization is like Example 2 on page 4. However, if one uses the randomization in Example 1, the cost is not decreased, but increased because of the extra multiplication. What is the benefit of introducing randomness at all?

- It would be more convincing if the paper shows some empirical results where the proposed algorithm brings a big improvement over the existing algorithm.



**Summary Of The Paper:**

This paper studies a primal-dual algorithm of a convex problem $f(x)+g(x)+h(Kx)$, where the update of the dual variable is randomized. The main result is the Theorem that says if $f$, $g$, and the conjugate of $h$ are all strongly convex, then the proposed algorithm RandProx converges linearly. The paper also proves the convergence rate for the convex case.

**Summary Of The Review:**

The paper studies an algorithm that simply adds randomization in one step of an existing algorithm. The motivation is not clearly stated and the improvement of the proposed algorithm is not well supported. The writing can also be improved with more examples and discussions.

Minor comments:
- Section 2.1 line 2: there should be a $\gamma$ in the definition of $\text{prox}_{\gamma\phi}$.

---

> ### Author Response · Authors · 2022-11-14
> **Reply to "Weaknesses"**
>
> > It is not clear why the problem $f(x)+g(x)+h(Kx)$ is important to study.
>
> This generic template covers essentially every problem of smooth or nonsmooth, constrained or unconstrained, optimization. Indeed, using a classical product space trick, as discussed in Appendix A.2, one can extend the formulation from 3 to an arbitrary number of functions. See for instance the tutorial paper "Proximal splitting algorithms for convex optimization: A tour of recent advances, with new twists" by Condat et al.
>
> Let us give 2 important application examples:
>
> 1) if $K=\mathrm{Id}$ and $h$ is the indicator function of the consensus constraint $x_1 = \ldots = x_n$, in distributed optimization, the proximity operator (prox) of $h$ is the projection on this constraint set, which corresponds to communication between the computing units. Thus, randomizing $h$ amounts to randomizing communication, which is the main bottleneck in distributed computing. This setting is discussed in Appendix A.3, see equations (27)-(29).
>
> 2) If $h$ is the indicator function of 0, then $h(Kx)$ corresponds to the constraint $Kx=0$, and randomizing $h$ also randomizes $K$. An application to decentralized optimization, where $K$ is the gossip matrix, is discussed in the ProxSkip paper by Mishchenko et al.
>
> > It is not clear why adding noise in the dual step is a good idea. It seems that the only benefit is that the cost per iteration might decrease if the randomization is like Example 2 on page 4. However, if one uses the randomization in Example 1, the cost is not decreased, but increased because of the extra multiplication. What is the benefit of introducing randomness at all?
>
> Randomizing a prox operator has virtually all the same benefits as randomizing a gradient operator in the wide class of SGD-type methods. For instance, SAGA and SVRG are methods to minimize a (possibly very large) sum of $n$ functions, which call at every iteration one gradient of a randomly chosen function instead of the gradients of all $n$ functions. The complexity in terms of number of iterations increases, but this is vastly compensated by the fact that the cost per iteration is divided by $n$. The exact same effect appears here with Algorithm 6 - RandProx-Minibatch, which has Point-SAGA as a particular case, as discussed in Appendix A.2. Point-SAGA is the counterpart of SAGA, with gradients replaced by proxs for the minimization of a sum of nonsmooth functions, for instance the hinge loss.
>
> > It would be more convincing if the paper shows some empirical results where the proposed algorithm brings a big improvement over the existing algorithm.
>
> > The paper doesn't have any numerical results supporting the applicability of the proposed algorithm.
>
>
> We acknowledge that our paper is theoretical, as we are laying the foundations of a framework for randomized proximal algorithms for nonsmooth or constrained optimization. In any case, we believe that theorems stating convergence to an exact solution, with tight bounds on the convergence rate, for instance the optimal rate for personalized federated learning given at the end of page 15, are far more convincing than a few plots. But we agree that it would be nice to illustrate our results by some examples. We can mention that our RandProx algorithm serves as the basis for a new algorithm with local training and compression in the paper "Provably Doubly Accelerated Federated Learning: The First Theoretically Successful Combination of Local Training and Compressed Communication" by Condat et al., Oct. 2022, where numerical results are given. They illustrate the large gain obtained by the combination of the two randomization processes of probabilistic activation and rand-k compression, which are two examples given in our paper; the third example of sampling is detailed in Appendix A.2.

---

> ### Author Response · Authors · 2022-11-14
> **Reply to concerns on "Clarity, Quality, Novelty"**
>
> > The algorithm proposed and studied by the paper is almost the same as the existing PDDY algorithm, except that the dual update is randomized.
>
> Yes, in the same way that SGD is almost the same as GD, except that the gradient is randomized. Joke apart, please note that our convergence rates are new even for the deterministic PDDY algorithm and its particular cases, like the Chambolle-Pock and Davis-Yin algorithms.
>
> > the paper doesn't give any practical scenario where the randomization is necessary or brings a significant improvement.
>
> We analyze in details several practical scenarios in Appendices A.1, A.2, A.3. Randomization brings the significant improvement of the convergence rate depending on $\sqrt{\kappa}$, which is optimal, instead of $\kappa$, the condition number, in several applications, see for instance at the end of page 15 and of page 18.
>
> > It seems that the paper is compressed from a much longer version to suit the page limit without careful checking. For example, Theorem 11 follows right after Theorem 4 while Theorems 5-10 are missing.
>
> We consider the Appendix as being fully part of the paper. All theorems are present. We give a broad perspective and the main messages on our approach in the main part of the paper, but the examples of applications are in the Appendix. We have 12 theorems and 16 Algorithms in total.
>
> > Equation (7) states that the variance of should be proportional to $|r^t|^2$. There is no discussion about what kind of randomization satisfies this assumption.
>
> This is a good suggestion and in the revised paper, we have added a paragraph after eq. (7). This assumption is satisfied by a large class of randomization strategies, which are widely used to define stochastic gradients. For Example 2 of probabilistic activation, $\omega = 1/p-1$, as written at the beginning of Section A.1. For sampling $k$ out of $n$ functions (Example 3 added in the revised paper) or selecting $k$ out of $d$ coordinates with the rand-k compressor (Example 1), the value of omega is $n/k-1$ (or $d/k-1$), as written after eq. 24.
>
> > Section 2.1 line 2: there should be a $\gamma$ in the definition of $\mathrm{prox}_{\gamma\phi}$.
>
> Yes, thank you for pointing out this typo. We have corrected it.
>
> > Correctness: 3: Some of the paper’s claims have minor issues. A few statements are not well-supported, or require small changes to be made correct.
>
> What are the issues in our statements and which changes are required to make them correct, wherever they are not? We believe that all our derivations are technically correct, please notify us if you have found any incorrectness.

---

> ### Author Response · Authors · 2022-11-18
> **Discussion?**
>
> Dear Reviewer,
> Did our replies and our modifications in the revised paper address your concerns? We don't think your overall and correctness scores of 3 are justified and we would be glad to discuss any point with you.

---

> > ### Comment · Reviewer_Lt4E · 2022-11-18
> > **Reply to author**
> >
> > Thank you for the reply and for revising the manuscript.
> >
> > My first concern is about the novelty of the proposed algorithm. I believe the benefits of using stochastic algorithms:
> > > SGD is almost the same as GD, except that the gradient is randomized
> >
> > >Randomizing a prox operator has virtually all the same benefits as randomizing a gradient operator in the wide class of SGD-type methods.
> >
> > After so many studies on SGD and its variants, however, I don't think proposing an algorithm where a single step is randomized can be considered novel. Even in the particular setting of minimizing the sum of three convex functions, there are a lot of stochastic primal-dual algorithms. Just to name a few:
> >
> > [1] Bianchi, P., Hachem, W. and Franck, I., 2014. A stochastic coordinate descent primal-dual algorithm and applications.
> >
> > [2] Yurtsever, A., Vu, B.C. and Cevher, V., 2016. Stochastic three-composite convex minimization.
> >
> > [3] Tran-Dinh, Q. and Liu, D., 2020. Faster Randomized Primal-Dual Algorithms For Nonsmooth Composite Convex Minimization
> >
> > So I don't think the algorithm is novel. Neither do I think the paper is properly positioned in the existing literature of randomized primal-dual algorithms.
> >
> > As the author argued, this is a theoretical paper.
> > > our convergence rates are new even for the deterministic PDDY algorithm and its particular cases,
> >
> > I acknowledge that the paper proves the linear convergence rates for deterministic PDDY algorithms. But please correct me if I'm wrong, I think the convergence rate for the deterministic algorithm is obtained by setting $\omega$ to 0. So the new rate for the deterministic algorithm can be proved without introducing randomization at all.
> >
> > > We acknowledge that our paper is theoretical, as we are laying the foundations of a framework for randomized proximal algorithms for nonsmooth or constrained optimization.
> >
> > Again, with all the existing literature, I hardly agree that this paper is "laying the foundation" for "randomized proximal algorithms for nonsmooth or constrained optimization".
> >
> > > We believe that all our derivations are technically correct, please notify us if you have found any incorrectness.
> >
> > I want to clarify that the correctness score of 3 is a reflection of my opinion that some statements in the paper are over-selling. For instance, "They are all generalized and unified within our new framework", "our generic algorithm RandProx paves the way to a new world of proximal counterparts of variance-reduced SGD-type algorithms". The framework of minimizing $f+g+h$ using a stochastic algorithm is not new. Also, it does not "pave the way" for all SGD-type algorithms. I'm willing to change it to 4 since I do not find any mathematical incorrectness.

---

> > > ### Author Response · Authors · 2022-11-19
> > > **Reply about novelty**
> > >
> > > Thank you for engaging in the discussion.
> > >
> > > We understand that your main concern is about the novelty of the conceptual mechanism we propose.
> > >
> > > > I don't think proposing an algorithm where a single step is randomized can be considered novel.
> > >
> > > > Neither do I think the paper is properly positioned in the existing literature of randomized primal-dual algorithms.
> > >
> > > > there are a lot of stochastic primal-dual algorithms. Just to name a few:
> > >
> > > [1] Bianchi, P., Hachem, W. and Franck, I., 2014. A stochastic coordinate descent primal-dual algorithm and applications.
> > >
> > > [2] Yurtsever, A., Vu, B.C. and Cevher, V., 2016. Stochastic three-composite convex minimization.
> > >
> > > [3] Tran-Dinh, Q. and Liu, D., 2020. Faster Randomized Primal-Dual Algorithms For Nonsmooth Composite Convex Minimization
> > >
> > > It is true that there exist randomized primal-dual algorithms, but most of them focus on randomizing the gradient. This is more difficult than in SGD, which consists in randomizing the gradient in the primal proximal gradient descent algorithm. But the "primal-dual" aspect is a difficulty which is different, if not orthogonal, to the difficulty of randomizing the proximity operator instead of the gradient.
> > >
> > > Indeed a gradient descent step is explicit, so that 'noise' in the gradient is additive, whereas the proximity operator is implicit, so that noise propagates in an intricate way. This can be seen in our randomized version (Algorithm 3 - RandProx-FB) of proximal gradient descent, which for $f=0$ reverts to a randomized proximal point algorithm. As we can see, even in the latter case, the algorithm keeps its primal-dual form and does not revert to something simpler and purely primal. In other words, the `noise mitigation' process is not straightforward. By contrast, in gradient descent, noise mitigation is essentially obtained by decreasing the stepsize. This is why we believe that our randomization mechanism is novel and significantly different from existing techniques.
> > >
> > > Let us also point out that in RandProx, there is underrelaxation on the dual variable to mitigate the noise on the dual variable, whereas the noise on the primal variable is mitigated by decreasing the stepzise $\tau$. Thus, our algorithm does not reduce to the type of  randomized Krasnosel'skii-Mann iterations or of stochastic quasi-Fejérian analyses present in the literature. We cannot reason in the abstract primal-dual product space and we have to enter into the details of the primal and dual steps. For instance, we don't currently know how to randomize $\mathrm{prox}\_{g}$ instead of $\mathrm{prox}\_{h^*}$ in the Chambolle-Pock algorithm (except of course by swapping the roles of the primal and dual problems, but then the algorithm is different).
> > >
> > > We agree that in the sampling case, RandProx can be viewed as a block-coordinate-type algorithm, with respect to dual variables. Thank you for reminding us this point. So, we will add references to stochastic coordinate-type primal-dual algorithms, which are indeed relevant. But our algorithm remains new, even in this setting, with the dual variables and functions activated randomly, whereas the 'primal' functions $f$ and $g$ are handled without any change. Yet, many questions are open, for instance on choosing the activation probabilities in a better way than 'uniformly at random' and for this the existing insights on coordinate-type methods will be precious.

---

> > > ### Author Response · Authors · 2022-11-19
> > > **Additional reply**
> > >
> > > > please correct me if I'm wrong, I think the convergence rate for the deterministic algorithm is obtained by setting  $\omega$ to 0. So the new rate for the deterministic algorithm can be proved without introducing randomization at all.
> > >
> > > This is correct, the deterministic algorithm PDDY is obtained by setting  $\omega$ to 0. This is actually by first deriving this better Lyapunov analysis for the PDDY algorithm that we gained insights on where the randomization mechanisms could take place and how the 'noise' could be mitigated. So, it is not that the randomized framework gives us insights even in the deterministic case, this is the other way around: a finer analysis in the deterministic case opened the door to randomization in the right way.
> > >
> > > > with all the existing literature, I hardly agree that this paper is "laying the foundation" for "randomized proximal algorithms for nonsmooth or constrained optimization".
> > >
> > > Indeed, our style is emphatic at some places, and certainly more than in similar papers in optimization journals, written in a neutral mathematical style. We believe that this is needed for theoretical optimization papers like ours to find their place in machine learning conferences, where they are perfectly legitimate, along with more applied papers promising, in a more or less subtle way, to change the world or save the planet. Our phrasing "laying the foundations" in not present in the paper, we used it in our reply. It was perhaps too strong and we apologize if this sounded arrogant. What we meant is that we consider our work as an important milestone for the design of randomized primal-dual algorithms.
> > >
> > > > I'm willing to change it to 4 since I do not find any mathematical incorrectness.
> > >
> > > Yes, please revise your score, since doubts about the significance of our work should not be confused with doubts about its correctness.

---

### Official Review · Reviewer_Q4Ce · 2022-10-24

**Confidence:** 4
**Correctness:** 4
**Technical Novelty And Significance:** 3
**Empirical Novelty And Significance:** Not applicable
**Recommendation:** 10

**Clarity, Quality, Novelty And Reproducibility:**

This paper is well written, very exhaustive with the particular settings the general framework can be applied to and the algorithms that (sometimes) it recovers.

The novelty is in the new framework and its generality. In many cases, this general framework recovers algorithms in previous works but they are seen from a single framework, with shared proofs. In some other cases, new and powerful results are obtained.

typos:

p.2 "which a proximal algorithm" ->

Second line of section 2.1, \gamma is missing in the definition of the prox

at almost the end of page 7 "with obtain the classical proximal algorithm" -> "we obtain the classical proximal algorithm"


**Strength And Weaknesses:**

This is a very general and clean framework that provides many algorithms and will help in the systematic analysis of other randomized primal-dual algorithms. Several of the applications lead to recover previous algorithms, but the novelty is in the general and simple framework, that allows for covering all those applications under the same analysis and for obtaining new results

In A.1, it is a bit of an odd (or loose?) assumption to assume that f_i are L_f smooth and then \sum f_i are L_f smooth, could you comment on that / clarify?

**Summary Of The Paper:**

Under strong convexity, this work provides a very general framework for primal-dual optimization, where the dual step, corresponding to a proximal operator, is randomized. Using a variance reduction technique, linear convergence rates are obtained. The optimizatio model is the sum of three convex functions, where the first one is smoth, while the other two are not, and the third one is the composition of a linear operator and a non-smooth convex function. Sometimes, this last function is assumed to be smooth, so the dual problem is strongly convex. A few results are provided in the case of just convexity of the objective. The work shows how this framework recovers many previous analysis in different settings and how new results are obtained.

**Summary Of The Review:**

As I said above, the paper is general, well-written and with interesting and powerful results.

---

> ### Author Response · Authors · 2022-11-14
> **Reply to the question on Lipschitz constant**
>
> Thank you for your very positive evaluation of our work. We have corrected the typos you noticed.
>
> > In A.1, it is a bit of an odd (or loose?) assumption to assume that $f_i$ are $L_f$ smooth and then $\sum f_i$ are $L_f$-smooth, could you comment on that / clarify?
>
> The only assumption is that every $f_i$ is $L_f$ smooth, for a value $L_f$ which is the same, for simplicity. Equivalently, every $f_i$ is $L_i$-smooth and we set $L_f:=\max_i L_i$. This implies that $f:(x_i) \mapsto \sum_i f_i(x_i)$ is $L_f$ smooth, in the product space $\mathcal{X}^n$. That is why we call this unique Lipschitz constant $L_f$. The fact that $f$ is $L_f$ smooth is not an additional assumption, it is a consequence of the smoothness of every $f_i$. Does this answer your question?

---

> > ### Comment · Reviewer_Q4Ce · 2022-11-15
> > **response**
> >
> > yes, I think I read $\sum f_i(x)$ instead of $\sum f_i(x_i)$ and that's why I was confused.

---

### Official Review · Reviewer_pXRx · 2022-10-24

**Confidence:** 4
**Correctness:** 4
**Technical Novelty And Significance:** 3
**Empirical Novelty And Significance:** 2
**Recommendation:** 6

**Clarity, Quality, Novelty And Reproducibility:**

The paper is well written and clearly explained. The work is original, however the novelty of the algorithm is limited in the sense that randomize existing deterministic methods with convergence guarantee is not a surprising result to date. No numerics were provided to verify the reproducibility.

**Strength And Weaknesses:**

Strength:
 - The proposed randomized algorithm can deal with the general three block composite optimization problem; several existing work in the literature can be casted as special case of the method.
 - For strongly convex cases, linear convergence rate was proved.

Weaknesses:
 - Practical side, lack of numerical examples to justify the advantage of the proposed scheme. In major parts stochastic methods, the randomization is applied to the gradient parts to reduce complexity. While for the non-smooth part, it is questionable whether randomization can bring as big advantage as the stochastic ``gradient'' methods. For example, when $h\circ K$ is total variation or wavelet like regularizations, $h^*$ accounts for simple projection, which is not necessarily of very high complexity.
 - When the problem is just convex, only convergence rate on the Bregman divergence of the primal variable $x$ was provided. However, the no rates for the dual variable. What caused this? Is it because that the dual function under this setting is non-smooth?

**Summary Of The Paper:**

The paper considered convex composition optimization whose objective function is the sum of three with one composed with a linear operator. A randomized algorithm, which generalized previous determined Primal-Dual algorithm, was proposed. Convergence rate in the strongly convex case was presented, while for the only convex case, convergence rate for the primal Bregman divergence was provided. Several special cases of the proposed algorithms were discussed.

**Summary Of The Review:**

The paper considered three-block composite optimization problem and proposed a randomized Primal-Dual algorithm to solve the problem. Convergence rates are provided, and discussions on some special cases are presented. No numerical experiments are provided to verify the advantages of the algorithms, which is a major drawback.

---

> ### Author Response · Authors · 2022-11-14
> **Response to "strength and weaknesses"**
>
> > Lack of numerical experiments.
>
> We acknowledge that our paper is theoretical, as we are laying the foundations of a framework for randomized proximal algorithms for nonsmooth or constrained optimization. As the saying goes, there is nothing more practical than a good theory! Our paper has 12 theorems and 16 algorithms covering different cases of practical interest, for instance Algorithm 9 - RandProx-FL for federated learning, see the discussion on why it improves upon the state of the art at the end of Appendix A.3. We did not take the time to run experiments to illustrate our findings, since we are focusing our efforts on investigating the many open questions raised by our new theory.
>
> > it is questionable whether randomization can bring as big advantage as the stochastic "gradient" methods. For example, when is total variation or wavelet like regularizations, accounts for simple projection, which is not necessarily of very high complexity.
>
> You are right, if $h$ is simple, like a $l_1$ norm or the indicator function of a simple set, there is no need to randomize the proximity operator (prox). But if the prox is expensive, it makes sense to replace it by a cheaper estimate, just like for the gradient.
>
> A promising venue is semidefinite programming where the variables are matrices and the functions are the nuclear norm or the indicator function of positive semidefiniteness, for instance. In that case, the prox requires a singular or eigenvalue decomposition, which scales like $\mathcal{O}(d^3)$ for a $d \times d$ matrix; this can be prohibitive. Strategies based on random sketching have been explored, see for instance "Scalable Semidefinite Programming" by Yurtsever et al., but this kind of approach is still in its infancy.
>
> We give 3 examples of applications:
>
> 1) skipping the prox; that is, the prox is computed only with a small probability, as discussed in Appendix A.1. An important application is distributed learning, where the prox (of the consensus contraint) corresponds to communication, which is the main bottleneck, and communicating only with a small probability captures the popular strategy of local training.
>
> 2) sampling; that is, only one prox or a small minibatch of proxs is computed at every iteration, instead of the proxs of all functions, as discussed in Appendix A.2. We recover Point-SAGA as a particular case, which is a direct counterpart to the SGD-type method SAGA for the minimization of a sum of nonsmooth functions, for instance the hinge loss when training classifiers.
>
> 3) compression. Several variants can be imagined. For instance, in a distributed for of the ADMM, the proxs would be computed but their output vectors would be compressed, using rand-k compressors for instance, to reduce the communication burden. Or the prox itself is replaced by a cheaper compressed form, like in the federated learning setting discussed in Appendix A.3.
>
> We have revised the paper and the example of sampling has been added as Example 3. It is still discussed in details in Appendix A.2.
>
> > When the problem is just convex, only convergence rate on the Bregman divergence of the primal variable x was provided. However, the no rates for the dual variable. What caused this? Is it because that the dual function under this setting is non-smooth?
>
> Yes, our study of the merely convex case is preliminary and we hope to derive stronger guarantees in the future. The analysis requires different tools than in the strongly convex case, with typically the Bregman divergence studied instead of the squared distance to the solution. Handling nonsmoothness, randomness, and the primal and dual variables jointly, is very challenging.
>
> > the novelty of the algorithm is limited in the sense that randomize existing deterministic methods with convergence guarantee is not a surprising result to date.
>
> On the contrary, we think that randomizing a prox within primal-dual algorithms is very new and an achievement, since this is quite harder than randomizing the gradient in the simple primal algorithm of gradient descent.

---

### Decision · Program_Chairs · 2023-01-20

**Decision:**

Accept: poster

**Justification For Why Not Higher Score:**

No reasonable numerical experiments.

It is understandable for not providing experiments for existing classic method that is included in this paper.

However, it is not proper for not providing experiments for the new results derived in this paper.

**Justification For Why Not Lower Score:**

Beyond providing some unified analysis for existing algorithms. The paper does provide some new algorithms and some improved convergence analysis for existing algorithms.

**Metareview: Summary, Strengths And Weaknesses:**

The paper considers a randomized variant of the Primal-Dual Davis-Yin algorithm, where the randomization is in the sense of randomly choosing the coordinate blocks to perform proximal oracles. Instead of proving the complexity of a specific randomization rule, the paper provides a set of general condition that enables the convergence. This makes it able to contain a variety of different methods and derive some interesting new results.

Overall, the paper is above the threshold for ICLR. However, there are a few issues that the authors should pay attention during the camera-ready revision.

(1). First of all, as the reviewer pointed out, some statements in the paper are over-selling. For instance, "They are all generalized and unified within our new framework", "our generic algorithm RandProx paves the way to a new world of proximal counterparts of variance-reduced SGD-type algorithms". The coordinately randomized algorithms are already very well-known in optimization. It will be better if the authors can revise the tone of these descriptions and fairly give the credits to the existing randomized coordinate methods.

(2). Second, the current description of the randomization is very vague. The readers will realize that the randomization is coordinate-wise for functions like $\sum_{i=1}^Nh_i(x_i), N\geq 1$ instead of the randomization for $\sum_{i=1}^Nh_i(x), N\geq 1$.  The authors may need to emphasize this when introducing the randomization operator $R(\cdot)$.




**Note From Pc:**

if the above contains the word "oral" or "spotlight" please see: "oral" presentation means -> notable-top-5% and "spotlight" means -> notable-top-25%. As stated in our emails, we are disassociating presentation type from AC recommendations

**Summary Of Ac-Reviewer Meeting:**

N/A